# Dynamic Analysis of the Almagrera Tailings Dam with Dry Closure Condition

Antonio Morales-Esteban [1,*], José Luis de Justo Alpañés [1], Pablo Castillo [1] and Muhammet Karabulut [2]

[1] Department of Building Structures and Geotechnical Engineering, Technical University School of Architecture, University of Seville. Av. Reina Mercedes 2, 41012 Seville, Spain; jlj@us.es (J.L.d.J.A.); pcastillo.iccp@gmail.com (P.C.)

[2] Department of Civil Engineering, Zonguldak Bulent Ecevit University, Incivez, Zonguldak 67100, Turkey; karabulut@beun.edu.tr

[*] Correspondence: ame@us.es

**Abstract:** In light of growing concerns over sustainability, particularly in the wake of environmental disasters like the Aznalcollar dam break, the Spanish authorities have heightened their awareness of issues surrounding ore tailings management. The main aim of this paper is to study the dynamic behavior for the dry closure of the Almagrera dam under the action of an earthquake. This study was carried out with the Plaxis 2D v9.02 program, which uses the finite element (FE) method. The dynamic analysis of the dam was interpreted in terms of deformations, displacements and principal stresses. The construction of the Uniform Seismic Hazard Acceleration Response Spectrum (USHARS) and the selection of real accelerograms for the time-history dynamic calculations is a noted feature of this research. Numerical analyses show that the dam is safe enough because a failure surface has not been formed, although several plastic zones may appear in the dam. The FE study of deformations display that the tailings may attain large deformations, displacements and failure, although this does not jeopardize the safety of the dam where the displacements are smaller than 3 mm. Neither the tailings nor the dam are expected to suffer liquefaction. It was determined that the 0.09 g threshold value is not exceeded in the acceleration-time graphs on the old reservoir field surface, which is the most critical situation.

**Keywords:** accelerograms; dam safety; dynamic calculation; earthquake analysis of dam; tailings dam; finite element; mining; Plaxis; sustainability

## 1. Introduction

Concern over the manner in which mineral waste sites are dealt with has increased recently, particularly after the catastrophic event of the Los Frailes dam in Seville, where there was a 4.5 hm³ toxic leak on 25 April 1998 [1,2]. As a result, in a changing new regime, mining companies have to propose a strategy for mine closure beforehand. Because of the insufficient insight in the former regulation, Spanish regional governments now have to overcome the challenge of the closure of numerous disused mines and mining tailings dams in the same manner as with the Almagrera mine waste fill dam.

According to the National Plan for the Waste of Extractive Industries [3], there are 610 tailings ponds and 378 tailings dams in Spain, with a total volume of 325,878,800 m³. All these tailings, most of them toxic, have been generated due to mineral processes. Andalusia, with a total volume of 153 million m³ (122 ponds and 4 tailings dams), has 47% of the total volume.

There are few studies related to the closure of tailings dams. [4] investigated the importance of capillary water considering one of the tailings dams existing in Cuba. A copper tailings dam in China was statically investigated [5]. Yin et al. [6] studied the impact of seepage on the structure of a mine waste dam. Various articles, contrarily, revealed the failures of tailings dams. Rico et al. [7] produced a detailed report on tailings dam failures

in Europe. Niekert and Viljoen [8] explained the reasons and outcomes of the Merriespruit dam problem, which is located in South Africa. Rico et al. [9] compiled the correlation hypothesis between geometry-related parameters in the tailings ponds and the hydrostatics properties of the overflow from historical mine waste fill dam failures. Gachechiladze-Bozhesku [10] analyzed the transboundary environmental impact assessment of the Neman Hydraulic Power Plant (Lithuania). Li et al. [11] studied in detail the earthquake analysis of reservoirs and huge dams after the paramount 8 Mw Sichuan earthquake [12,13]. Costa-Pierce [14] examined the sustainability of constructing new hydropower reservoirs in relation to the relocation of the population and cage aquaculture.

In contrast with the few examples related to the closure of tailings dam, there are many examples connected with the dynamic analysis of earth-dams stability [15–28]. It is important to note that, traditionally, quasi (pseudo)-static approaches were utilized to assess fill dams in statical terms. This pattern includes lots of simple presumptions. The analysis of dams is a complex geotechnical issue which needs dynamic analysis. An appropriate characterization of the materials, an accurate formulation of the problem, and a suitable stress–strain soil behavior model are required in a dynamic calculation.

Static and pseudo-dynamic analyses of the rockfill dam were examined for different reservoir water levels using the finite element method [29]. When pseudo seismic analysis and static cases are compared, the displacement of the dam body in the full reservoir case is approximately twice that of the static analysis. This situation reveals the importance of seismic analyses in terms of displacements. In addition, the earthquake performance analysis of dams and dams behavior under the galleries in the dam bodies were investigated by earthquake analysis using the finite element method [30].

Improving tailings dam safety through soil treatment was investigated using the Plaxis program, taking into account the phased construction situation with the finite element method [31]. The numerical model showed that the use of cement kiln dust, re-cycled gypsum B mixture, increased the overall stability of the tailings dam above the traditional 1.5 factor of safety requirements. Failure of tailings dams under different conditions was investigated [32]. The results display that the grain size distribution of the model sand should be moderate. The composition of the particle size distribution has a significant impact on the collapse morphology of the dam after failure. An experimental and theoretical study with the Tailings Dam with Geotextile Bag (TDGB) was carried out using the stability analysis method based on the limit equilibrium theory of tailings dams [33]. With respect to the test and analysis results, TDGB has displayed enhanced stability compared to the general tailings dam. An empirical function has been validated concerning the risk assessment by a simulation of tailings dams failure [34]. A dam located in the Ledong region of China was chosen as an application model and the static and seepage analyses of tailings dams were studied with the finite element method [35]. According to the outcomes, the seepage deformation failure of the main dam and the auxiliary dam occurred in neither case (normal water level and high-water level). To perform time-domain dynamic analysis of tailings dams, a power spectrum-based approach was studied by scanning seismic records [36]. An excellent fit is achieved with displacements and a reasonable one with pore water pressures in a dam using Plaxis 2D [37]. The measurement values obtained with a device and the Plaxis 2D finite element model results were quite consistent. This validates the suitability of using the Plaxis 2D program in similar studies. Numerical analysis of innovative seismic response of earth-rock dams studies were carried out with antiseepage walls [38]. The dynamic viscoelastic constitutive model was utilized in this research. The buckling fracture mechanism of broken rock dam foundation was investigated by gentle transitions and vertical structural discontinuities [39]. The modeling philosophy and process, and outcomes for the rock dam foundation are defined and prompted by using numerical methods. A buckling type of failure mechanism is confirmed by analyzing the deformation properties resulting from the overloading of the strength reduction of the numerical method. A comprehensive diagnostic method for the safety of tailings dams based on the dynamic weight and quantitative index was studied [40]. For the deformation

stability project, the amount and rate of deformation are determined by analyzing and interpreting normal operating data in situ observational data and combining them with numerical simulation outcomes.

The novelty of this research is that the dynamic analysis of the Almagrera ore tailings dam is studied for the dry closure situation. The fact that this situation is unique and that this issue has never been studied concerning the Almagrera dam before makes this study more valuable. After being informed of the current status of the mine waste fill dam, dynamic analysis was conducted to evaluate the dam static balance. Very few works describe dynamic balance analyses for mine waste fill dams. Execution of the dry closure of the Almagrera mine waste fill dam is currently almost finished, although some final operations are still required [41]. A dynamic calculation, introduced in a previous publication [42] is developed here. Initially, the probabilistic hazard equation for the site has been figured out. According to the hazard graphs gathered, the Uniform Seismic Hazard Acceleration Response Spectrum (hereinafter, USHARS) was produced for the position, based on the requested hazard rating (possibility of failure and exposure time) and the soil type. Calculation accelerograms were determined afterwards. According to this technique, exact accelerograms were gathered for a 1000 years reiteration duration. Later, the dynamic parameters were obtained. Finally, the dynamic calculation was performed with Plaxis 2D software (v9.02) and the results interpreted. Three sections of the dam were studied via 2D analysis. An analysis of the liquefaction susceptibility for the site was also included.

## 2. Location and Geology of the Almagrera Tailings Dam

The Almagrera region is positioned near the village of Calañas, in Andalusia (Huelva), east of the province and southwest of Spain. Aznalcóllar, situated to the east of Almagrera, is nearby.

Geologically, the area is located in a region known as Faja Piritica [43], which is one of the three divisions into which South Portugal is separated. The Pyrite Belt has sedimentary and igneous rocks of the late Devonian–Carboniferous age. Further, it consists of three different litho-stratigraphic units. Emerging from the late Famennian and Frasnian periods, quartzites and phyllites are older than others [44]. The Pyrite Belt, which is likely to include the greatest accumulation of volcanogenic stupendous sulfide bed worldwide, is one of the most remarkable provinces known of. Over 80 mineral beds of sulfurs and 300 of magnesium are situated within it. As a result of this situation, a very long period of mining activity has taken place in this region.

The geotechnical properties of the dam materials are given in Table 1 (in regard to in situ laboratory experiments, extracted from research [41] with some minor corrections).

The maximum height of the Almagrera dam is 37.3 m. Due to the accumulating material on its downstream side, it was lifted approximately five times. The side incline was 1.7 (H): 1(V) in the initial three times and 2 (H): 1 (V) in the final two times. The most unfavorable cross-section is given in Figure 1 before the closure and after five raisings. The fill dam has an upstream sloping center zone. A succession of magmatic and inter-stratified deposit rocks shaped its foundations, containing shale, phyllite, lava, and clayey. On the fifth rising period, an aslope gravel filter and sand was set among the fourth and fifth ascending covers, performing a filter standard, in addition to a downstream bottom discharge down it and the downstream cover. As much as a 16 m$^3$/h leakage appeared in the downstream incline. The products of the excavations that were obtained were placed in the dam, but with rockfill at the base. There were also different drains (in yellow) and quarry-run (permeable) on the outside. The unfavorable section plan detail of the Almagrera dam before its closure is presented in Figure 1.

**Table 1.** Calculation parameters of the dam materials. Fifth raising.

| Description | Classification (USCS) | $c'$ (kPa) | $\Phi'$ (°) | $\gamma$ (kN/m$^3$) | K (m/s) | E (MPa) |
|---|---|---|---|---|---|---|
| Clayey and silty slate. Quarry run. | GC | 6 | 33 | 20.0 | $6.5 \times 10^{-5}$ | 30 |
| Rockfill | SC | 15 | 31 | 21.9 | $9.5 \times 10^{-7}$ | 60 |
| Sand filter. 5th raising | SP-SM | 1 | 35 | 20 | $10^{-5}$ | 50 |
| Quarry-run (weathered schist). 5th raising | GC | 6 | 33 | 20.2 | $6.5 \times 10^{-5}$ | 30 |
| Sound rock | | 250 | 20 | 21.4 | $1.3 \times 10^{-7}$ | $10^4$ |
| Tailings (average) | ML | 8.8 | 31 | 28.8 | $1.4 \times 10^{-8}$ | 2 |
| Clay core. 5th raising | SC | 18 | 30 | 19.8 | $10^{-8}$ | 50 |
| Sand filter | SP-SM | 1 | 35 | 20 | $10^{-5}$ | 50 |
| Selected rockfill | | 1 | 35 | 20 | $5.1 \times 10^{-3}$ | 60 |
| Clay core | SC | 18 | 30 | 19.8 | $10^{-8}$ | 50 |
| Weathered rock | SC | 50 | 20 | 20.5 | $1.4 \times 10^{-6}$ | 300 |
| "Las Viñas" fill | CL-ML | 32 | 26 | 22.3 | $5 \times 10^{-7}$ | 12.6 |

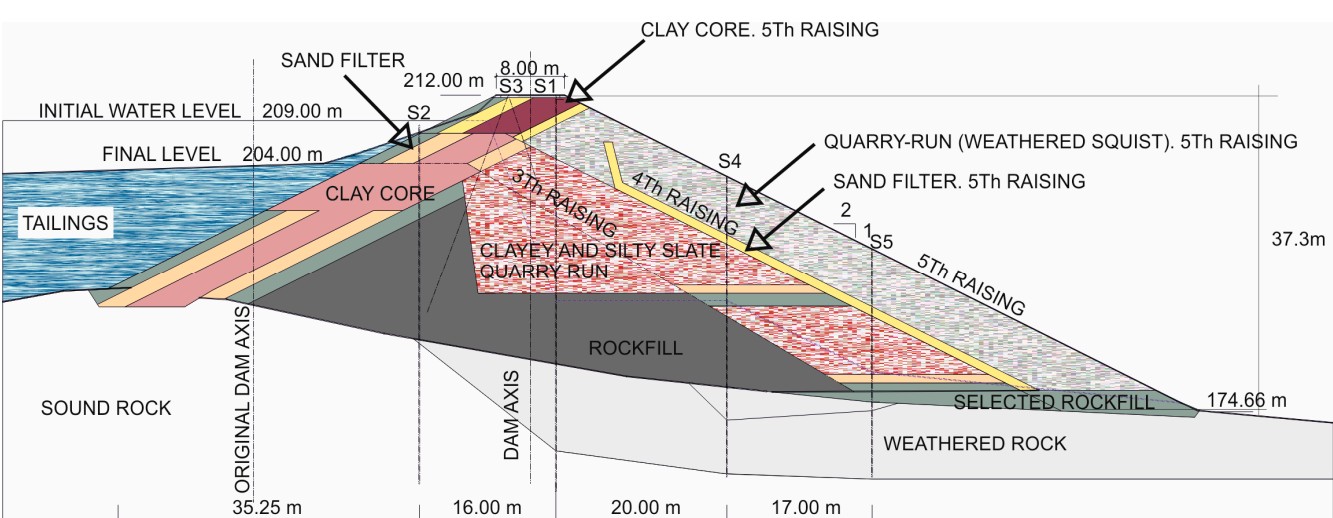

**Figure 1.** Unfavorable section plan detail of the Almagrera dam prior to closing down (S = borehole), adapted from [41].

## 3. The Dry Closure

A plan for the dry closure of the mine was promoted by the Ministry of Science and Innovation of the Spanish Government. The dry closure of a dam means that all the water accumulated in its tailings is going to be extracted and that it is going to be waterproofed to avoid water entering in the future. A brief description of the project follows. The details of the dry closure are presented in the flow chart in Figure 2 below for a better understanding.

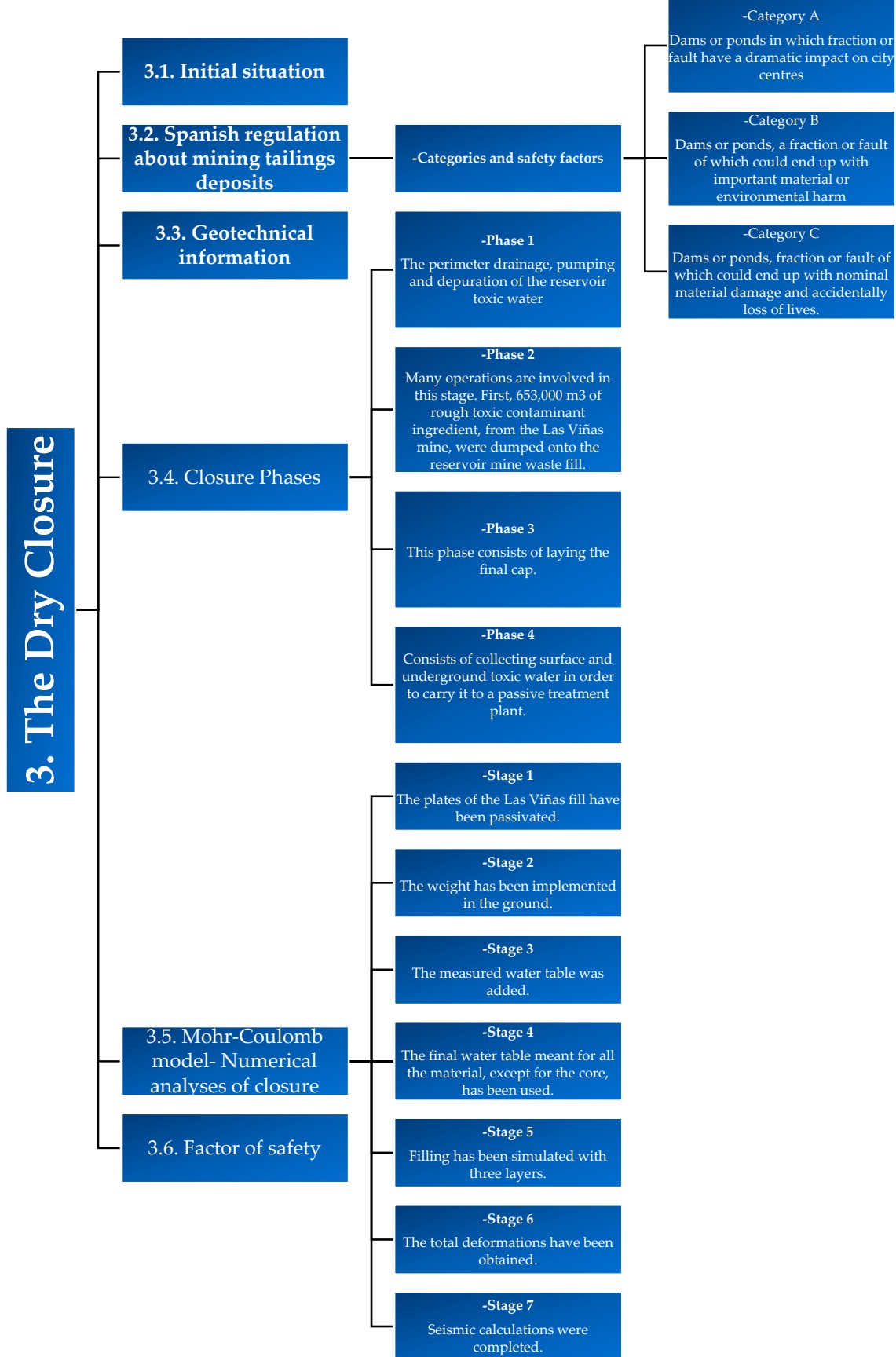

**Figure 2.** Flowchart of dry closure stages of Almagrera dam.

### 3.1. Initial Situation

At the first stage of the project, a great deal of toxic accumulation occurred a few meters underwater. The puddle was absolutely polluted by the chemical mine waste fill and rose two hundred and nine meters above sea level (Figure 1). During the closure works (see Section 3.4), the original section was modified by a rockfill reinforcement. In addition, a drawdown was carried out and more material (the Las Viñas fill) was put above the tailings. Figure 3 displays the final central section plan detail.

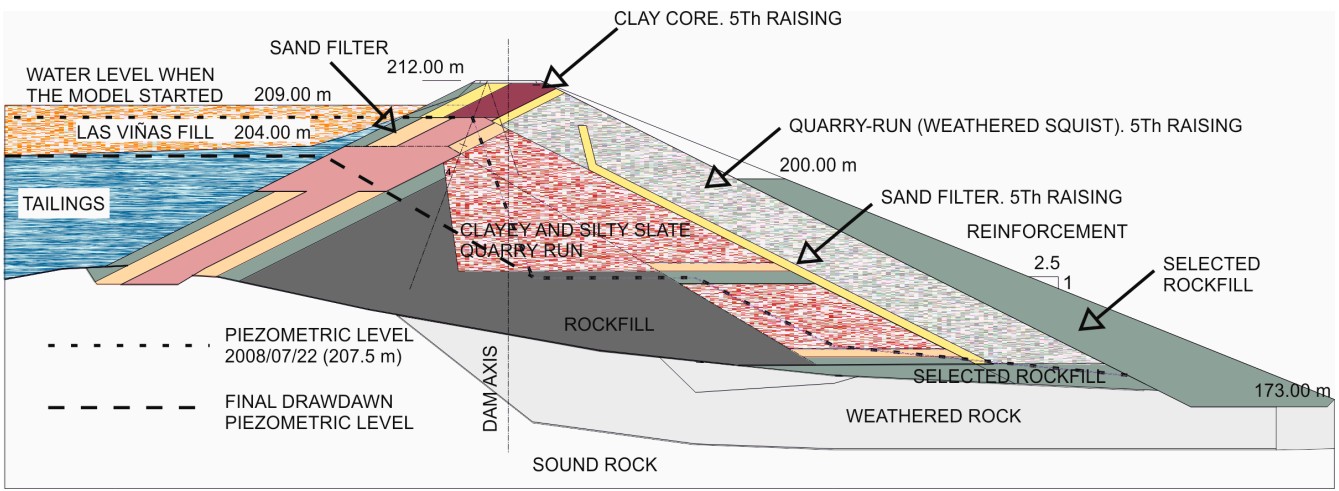

**Figure 3.** Final central cross-section, adapted from research [41].

### 3.2. Spanish Regulation about Mining Tailings Deposits

It was essential to recur to the rules for dams since the current Royal Decree [45] on the handling of mine waste fill and rehabilitation and protection of the area influenced by mining operations does not provide certain rules for reliability [46].

Categories and Safety Factors

In the safety standard stipulated by previous research [47] on categorization, ponds and dams are categorized based on the possible losses stemming from their suppositional fraction, or their fault, in Category A, B or C. In all events, Category C contains entire rafts and dams not counted in Categories A or B. However, the Royal Decree 975/2009 states the subsequent cases for a waste plant to be categorized within Category A:

(a) If, in compliance with a hazard evaluation, a crucial incident may stem from a failure or malfunction; for instance, the demolishing of a mass or the fraction of a dam.

(b) If it comprises substances categorized as hazardous under Directives 91/689 or 67/548/EEC over a definite level.

Three types of movements are taken into account with respect to their hazard and constancy: ordinary movements, incidental movements, and extraordinary movements. Ordinary movements are permanent movements: hydraulic compression, pore water compression, weight, and mine waste fill compression. Incidental movements are restricted period movements: e.g., strong ground motions with a predictable intensity or swift drawdown. In the end, actions are classified as extraordinary when they only occur under extreme conditions. Finite element (FE) methods and limit equilibrium (LE) approaches are used for computation. Minimum SFs are given in Table 2. Once the finite element model was performed, a consulting firm categorized the Almagrera fill dam in Category C. Subsequent investigations [48] have concluded that "the supposed toxic spill would have a dramatic (environmental) impact, like the one that occurred in 1998 at Aznalcóllar, an abandoned mining site showing strong similarities with Almagrera". With respect to the article, it was supposed to be included in Category A. Once the consequences of the computation are analyzed, that issue will be reconsidered.

**Table 2.** SFs stipulated by the Spanish regulations (Ministerio de Agricultura, Pesca, Alimentación y Medio Ambiente, 2011b).

| Type of Movement | Dam Category | |
|:---:|:---:|:---:|
| | **A or B** | **C** |
| Ordinary | 1.5 | 1.4 |
| Incidental | 1.3 | 1.2 |
| Extraordinary | 1.1 | >1.0 |

### *3.3. Geotechnical Information*

In situ and laboratory experiments have been executed to characterize the properties of the dam materials. Five boreholes—one on the top, two in the downstream slope, and two in the upstream slope—have been bored in each of the three cross-sections analyzed, spaced about 50 m apart. The distribution of boreholes in the central section can be observed in Figure 1. Table 1 shows the materials considered and their calculation parameters. It should be noted that soft tailings and medium tailings are two hypotheses considered for the same material.

### *3.4. Closure Phases*

The operations in the Almagrera tailings dam, in terms of the dry closure, started on 29 May 2008. The closure can be divided into four phases:

#### 3.4.1. Phase 1

This phase has two stages. The first one was the perimeter drainage and the second was the pumping and depuration of the reservoir toxic water before discharging it into the river.

#### 3.4.2. Phase 2

Various work is involved in this stage. First, 653,000 m$^3$ of a rough toxic contaminant ingredient from the Las Viñas mine was dumped onto the reservoir mine waste fill. Second, the outside was leveled and a 10 cm deep clayey plate was put on the top. In certain places in which the mechanism had trouble riding the mine waste fill, a geotextile was set on the surface. Next, the drainage wells were constructed. Finally, the dam reinforcement was executed.

#### 3.4.3. Phase 3

This phase consists of laying the final cap. It has not been started yet.

#### 3.4.4. Phase 4

The last phase will consist of collecting surface and underground toxic water in order to carry it to a passive treatment plant.

### *3.5. Mohr-Coulomb Model-Numerical Analyses of Closure*

Numerical analyses of closure were conducted via the Finite Element (FE) method and the triangular 15-nodes elements utilized. To analyze the materials performance, a Mohr-Coulomb formula was utilized, which is a pattern of perfect, non-associated plasticity. The six yield functions (1) describe a hexagonal shape in the stress region [49]. Mohr-Coulomb failure surface in principal stress space is depicted in Figure 4.

$$f_{ij} = \frac{1}{2}\left(\sigma'_{ic} - \sigma'_{jc}\right) - \frac{1}{2}\sin\Phi'\left(\sigma'_{ic} + \sigma'_{jc}\right) \leq 0 \tag{1}$$

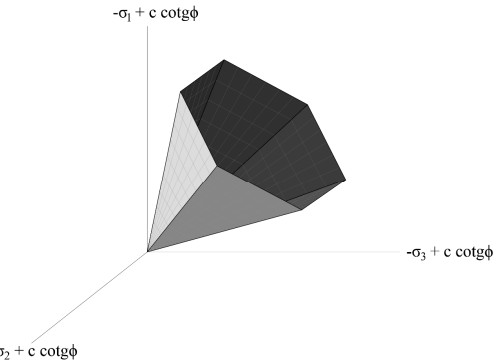

**Figure 4.** Mohr-Coulomb failure surface in principal stress space.

Being $\sigma'_{ic} = \sigma'_i + c\prime \cot \Phi\prime$The possible six plastic functions:

$$g_{ij} = \frac{1}{2}\left(\sigma'_i - \sigma'_j\right) - \frac{1}{2}\sin \Psi \left(\sigma'_i + \sigma'_j\right) \leq 0 \tag{2}$$

In both expressions i = 1, 2, 3; j = 1, 2, 3 (I $\neq$ j).

Stress circumstances within the yield surface have elastic behaviors, and the whole strains are reversible. The yield functions (1) are governed by the two classic factors: friction angle ($\Phi$) and cohesion (c). Conversely, the possible plastic functions include a third plasticity variant, which is the dilatancy angle ($\Psi$). The variant models positive plastic volume increase or dilatancy, especially monitored for intense grounds. For c > 0 the model allows for tension, the following being the functions which define this behavior.

$$f_{kt} = \sigma'_k - \sigma'_t \geq 0 \tag{3}$$

where k = 1, 2, 3.

When needed, the dilatancy angle is calculated according to the Plaxis recommendation:

$$\Psi = \Phi' - 30^{\circ} \tag{4}$$

Numerical analyses of closure have consisted of the following stages:

### 3.5.1. Stage 1

First state. The plates of the Las Viñas fill have been passivated. The initial water level has been considered and the materials have been considered drained.

### 3.5.2. Stage 2

Gravity. The weight has been implemented to the ground. Materials have been considered drained.

### 3.5.3. Stage 3

Drawdown to the 207.5 m level has been considered. The water table measured on the 22nd July 2008 is added. The materials have been considered drained.

### 3.5.4. Stage 4

Total drawdown. Two calculations have been applied:

(a) The final water table meant for all the material, except for the core, has been used. For the core, the initial water table reducing the water height over the upstream face has been estimated. The materials have been consolidated, initially undrained. As the total drawdown is supposed to be applied at the beginning, the time has been reduced to half (45 days).

(b) From the initial water table the reservoir level reduction has been applied in the upstream face as a transient regime for 90 days through the PlaxFlow program.

### 3.5.5. Stage 5

Fill. This has been simulated with three layers of similar thickness for profiles 1 and 2. Two layers have been considered for profile 3. As every layer is instantly applied, consolidation times of 72, 36, and 12 days have been used for three layers and 72 and 42 days, for two layers.

### 3.5.6. Stage 6

Long term stability. From phase 2, considering the fill and the drawdown water table. The total deformations have been obtained. Also, two hypotheses have been considered:

(a) Calculation with piezocone level.
(b) Calculation with flow net.

### 3.5.7. Stage 7

Seismic calculations, including considerations for tailings liquefaction.

### *3.6. Factor of Safety*

According to the Spanish regulation (Table 2), the Almagrera tailings dam is included in Class 1 due to its height and in Category C (moderate damage). Considering the dam class, the safety factors contained in the first row of Table 3 must be fulfilled. In order to fulfil these conditions, compacted rockfill reinforcement has been projected, which may be observed in Figure 2. Table 3 displays the outcomes of the pseudo-static calculation with the Finite Element Method (FEM) [41]. The phases described in Section 3.5 have been considered. The phi/c reduction procedure [50] has been used to calculate the factor of safety. It must be noted that it is not necessary to reach the safety factors inside the tailings, since this would imply a tailings movement and not a dam failure. The values indicated in the table correspond to the minimum safety factor and the largest displacement for hypothesis a and b in the total drawdown of the closure phases.

**Table 3.** Safety factors and final displacements with the FE method.

| Phase | δmax (mm) | Factor of Safety |
|---|---|---|
| First | | 1.44 |
| Reservoir at 207.5 level | | 1.44 |
| Drawdown | 350 | 1.34 |
| Fill | 1290 | 1.98 |
| Long term | 1610 | 2.74 |
| Long term extreme | 1730 | 1.31 |

## 4. Seismicity and Seismic Input Data

To understand the dynamic activity at the Almagrera location, it is important to recognize the seismicity in the Iberian Peninsula and its causes, which are herein briefly described. Then, the necessary seismic input data are calculated, first as required by the Spanish regulation, second as is needed for the dynamic calculation.

### *4.1. Seismic Properties of the Iberian Peninsula*

The contiguity directed NWSE between Eurasia and Africa, along that region of the boundary between plates, causes the deformation of the shell of the Iberian Peninsula, the Maghreb, and the neighbor coastal regions of the Atlantic and the Mediterranean [51–53]. The layers' boundary is heterogeneous, with sequential tectonic and oceanic regions bonded and gradual alterations in the stress direction. The region, with respect to the northwest of Africa and the Iberian Peninsula, may be regarded as the most complex connection region,

with an un-extreme seismicity coupled with the magnitude of strong ground motions. That region is confined with a frequent seismic activity with destructive strong ground motions on both sides. The coexistence of wide and compressive tectonics, and the interaction of the Iberian micro-plate [54,55] makes understanding the region extremely complex.

### 4.2. Seismicity at the Almagrera Site

The Almagrera tailings dam belongs to the South Portuguese unit. This area is classified by thrust faults with a NW-SE orientation in a compressive regime [56]. The maximum expected magnitude is 5.5, the lowest value in the Iberian Peninsula. The annual rate of earthquakes is also low (about $1.6 \times 10^{-4}$/km$^2$ for $M_w \geq 3.0$). Thereby, it can be concluded that the Almagrera tailings dam is not situated in a critical seismic zone, but the seismic risk of the zone is affected by very seismically active zones such as the Azores-Gibraltar fault.

### 4.3. Spanish Regulation

The Spanish regulation that states the criteria and calculation methods for seismicity is called Norma Sismorresistente, 2002–Earthquake-resistant Construction Regulations-(NCSE-02). This regulation provides a seismic hazard map with basic acceleration ($a_b$) isolines. In addition, it supplies a list of the basic seismic accelerations for every town or city. For Calañas village $a_b = 0.08$ g.

According to the regulation, the seismicity for this basic acceleration is:

Medium: 0.04 g < $a_b$ < 0.13 g. Pseudo-statics methods can be used. Technical Guides (2 and 3) of the Spanish Committee on Large Dams (SPANCOLD) consider constant forces acting on the center of gravity. The horizontal component is produced by the basic acceleration; the vertical one is reduced by 0.7. Basic seismic acceleration has to be modified depending on the specific case. The NCSE-02 method to obtain calculation acceleration is:

$$a_c = S \cdot \rho \cdot a_b \tag{5}$$

where:

- $a_b$: basic seismic acceleration.
- $\rho$: dimensionless coefficient which takes into account the return period. For constructions of special importance this is 1.3—which means a return period of 1000 years.
- $S$: soil amplification coefficient.

$$S = \frac{C}{1.25} + 3.33\left(\rho \cdot \frac{a_b}{g} - 0.1\right)\left(1 - \frac{C}{1.25}\right) = 0.803 \quad \text{for } 0.1 \text{ g} < \rho \cdot a_b < 0.4 \text{ g}$$

- $C$: Soil coefficient depending on the classifications of the soil (Table 4).

where $V_s$ represents the shear wave velocity with respect to the foundation. For multilayer foundation soils, the first 30 m should be taken into consideration. In this case, $C$ is calculated as a weighted average:

$$C = \frac{\sum_i C_i \cdot e_i}{30} \tag{6}$$

where $e_i$ is the layer thickness.

**Table 4.** Soil coefficient.

| Type of Soil | Characteristics | Coefficient $C$ |
|:---:|:---:|:---:|
| I | $V_s > 750$ m/s | 1.0 |
| II | $400 < V_s < 750$ m/s | 1.3 |
| III | $200 < V_s < 400$ m/s | 1.6 |
| IV | $V_s < 200$ m/s | 2.0 |

At the location of the Almagrera dam:
$a_b$ = 0.08 g (Provided by NCSE-02)
$\rho$ = 1.3 (Special importance, $T_r$ = 1000 years)
$C$ = 1.0 (Basically sound rock)
So, $a_c = S \cdot \rho \cdot a_b$ = 0.803·1.3·0.08 g = 0.083 g
The International Commission on Gigantic Dams advises, for Category C dams, a repetition time of 1000 years and the design acceleration ($a_d$):

$$a_d = 1.3, \text{ so, } a_b = 0.104 \text{ g} \tag{7}$$

The acceleration was embraced in the finite element model. For the *design earthquake* in dams of Category A, this kind of design acceleration was carried out. For Category A dams, the Spanish legislation advises a destructive strong ground motion as well, considered as an excessive movement, by design acceleration:

$$a_d = 2, \text{ so, } a_b = 0.16 \text{ g} \tag{8}$$

Regarding Table 2, seismic forces should be calculated as accidental actions; therefore, the safety factor should be at least 1.2. The consequences of the pseudo-static calculations are listed in Table 3. The lowest safety factor is 1.34, obtained for the drawdown phase (>1.2). A safety factor of 1.03 has been calculated inside the tailings—not in the dam—and for this reason the dam is considered safe enough. It is important to note that a dynamic calculation for Almagrera is not required by the Spanish regulation. However, considering that the Almagrera tailings dam includes the largest mine waste deposit in the Andalusian region, a dynamic calculation is appealing. Large displacements occur inside the tailings.

*4.4. Definition of the Accelerograms*

The initial step in a seismic analysis is the definition of the accelerograms that will be applied. Where many accelerograms have been recorded for years, accelerograms near the site could be used. By contrast, where there are not so many data available, using visco-elastic response spectra is usually a better option. The probabilistic method for choosing calculation accelerograms proposed by [56] has been used. This method is grounded on the construction of a USHARS for the site, considering the kind of ground and the desired risk situation—time of exposure and possibility of exceeding it. Then, the USHARS is compared with the acceleration response spectrum of real accelerograms recorded in the identical kind of ground. Finally, the accelerograms with a scale factor (f) close to 1 (Table 5) and a minor standard declination have been chosen.

**Table 5.** Selected accelerograms for the Almagrera mine waste fill dam.

| Accelerograms | f |
|---|---|
| 358 | 1.08 |
| 385 | 0.915 |
| 607 | 1.043 |
| 4341 | 1.006 |
| 6261 | 1.050 |
| 6269 | 0.964 |
| 6274 | 0.923 |

For the Almagrera location, the following features and parameters have been considered:
-    Founded on sound rock.
-    Repetition time shut to 1000 years.

Real accelerograms were obtained at http://www.isesd.hi.is/ (access date: 4 December 2023). Data is available in the European Strong Movement Database, available online at

this online address. The accelerograms selected are displayed in Table 6. Figure 5 displays the USHARS for the site and compares it with the spectrum of selected earthquake records.

**Table 6.** $AES_{min}$ and $AES_{model}$ in each case. M-C = Mohr-Coulomb model. HS-S = HS-Small model.

| Section | S1. M-C. | S1. HS-S. | S2. M-C. | S2. HS-S. | S3. M-C. | S3. HS-S. |
|---|---|---|---|---|---|---|
| $V_s$ (m/s) | 1737.70 | 1737.70 | 2822.29 | 2822.29 | 1782.38 | 1782.38 |
| $f_r$ (Hz) | 25 | 25 | 40 | 40 | 28 | 28 |
| $AES_{min}$ (m) | 8.69 | 8.69 | 8.82 | 8.82 | 7.95 | 7.95 |
| $AES_{model}$ (m) | 5.39 | 2.84 | 6.29 | 3.31 | 5.33 | 3.99 |

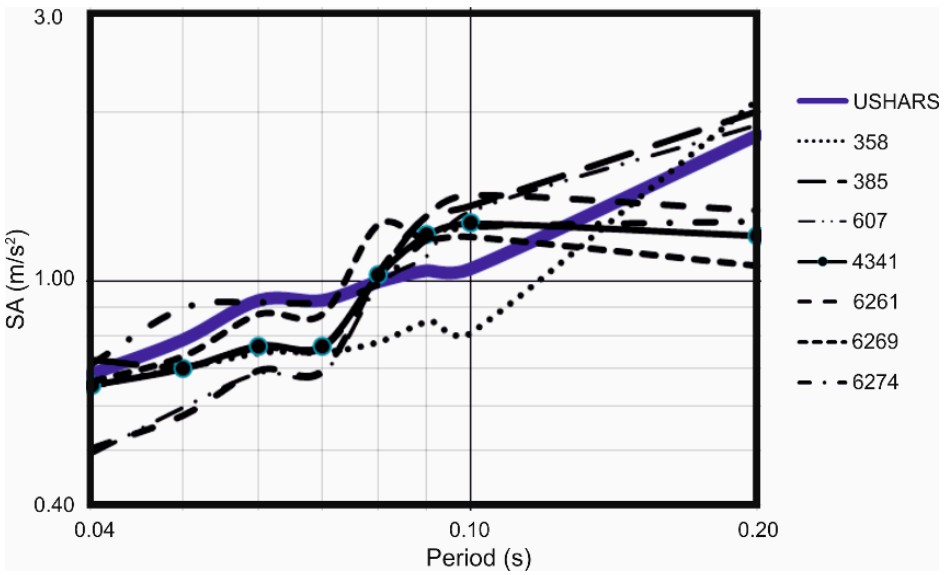

**Figure 5.** Comparison between the USHARS and the earthquake response spectra of the accelerograms chosen.

## 5. Dynamic Calculation

Once the accelerograms have been chosen, the dynamic computation is carried out for the Almagrera tailings dam. Three sections spaced about 50 m apart have been calculated.

### 5.1. The Safety of Dams during Earthquakes

Pseudo-static methods are still used by a great number of stability analyses, in which lateral forces are utilized in place of accelerograms. From the paper written by various researchers [57] the dynamic seismic computations were built upon the presumption that the hysterically stress–strain behavior of the ground can be established by a corresponding elastic method of linear analysis, founded upon a damped elastic model linearly when the characteristics of that finite element model were convincingly selected. This model is called Equivalent linear model.

An earthquake researcher already realized that the ordinary theory of a Factor of Safety (FOS) on shear strength does not thoroughly evaluate the response of an incline when large seismic actions occur. The FOS is the parameter by which the strength ought to be decreased to obtain the incline in a circumstance of restriction balance with all the stresses throughout a failure upper part. A section of the slope will slide throughout the failure surface when the FOS is smaller than one. This condition should only be maintained under static circumstances because stresses appear until large ground deformations change the structure shape. In the case of seismic circumstances, on the other hand, this is likely to lead the FOS to reduce momentarily to less than one, because this situation will continue for a while and the incline may stop when the latest stresses are not more than the current

strength. Performance should be evaluated according to the relative deformations to which the sliding part may be subjected when a strong ground motion occurs.

The permanent deformation stemming from strong ground motions cannot be calculated through this kind of analysis as the possible analysis with strain–conformable ground characteristics in the same way as in a linear numeric model is completely elastic. Nevertheless, it is argued that the stresses resulting from these strains represent the stresses in the ground and the accelerations represent the area values in an acceptable manner [58]. In order to estimate earthquake-induced permanent displacements in dams, earthquake dam deformation models are utilized.

The rigid-block model was the first model developed by previous research [59]. This model was modified to comprehensive sliding through a sloping plane by various researchers [60], who initially investigated the statical balance of a stiff mass reclining on a plane face. The acceleration angle has a minor effect on the decisive result [60]. In a dam, the first requirement is to choose the crucial slip surface which will provide the minimal key acceleration $K_{cg}$, in a pseudo-static computation. Ref. [61] suggested a method beginning with the specifying of the sliding face and the critic (yield) dynamic coefficient, ky, with the periodic yield strength, like in Sarma's approach. If the induced acceleration goes beyond the computed critical acceleration for a particular possible sliding mass, seismic activities are supposed to take place through the significance of the deformation and the way of the failure surface is asserted through a dual integration operation. Once yielding happened, the way of action for a possible sliding block was supposed to be through a lateral plane. These authors show a facilitated operation for predicting the natural period and the peak crest acceleration, $T_0$, of a barrier connected with a specific base action [62]. They identify the "maximum average acceleration", $k_{max}$. They gather one graph relating $\frac{u}{k_{max}gT_0}$ with $k_y/k_{max}$ for distinct values of Magnitude thanks to a number of earthquake records. That chart would provide the predictive ultimate displacement, u.

Earthquake FE analyses may be regarded as the most complete tool for estimating the dynamic behavior of a geotechnical technique, because they provide comprehensive knowledge of the soil deformation and stress distribution [63]. Nevertheless, a suitable soil constitutive model, a reliable soil characterization and a correct definition of the seismic input are absolutely necessary. Figure 6 displays the three FE models.

In addition, several parameters responsible for numerical damping are very important. The damping corresponding to the model in an FEM depends upon the main model (material damping), the borderline circumstances, and the integration scheme of the formulas (numerical damping). The calculations have been performed under undrained conditions which is appropriate for a dynamic analysis.

Cross sections of different unfavorable regions were taken into account. In this way, the dam is interpreted not only for one section but for three different sections: section 1, section 2 and section 3, and is presented in Figure 6. Charatpangoon et al. [64] have investigated the collapse behavior of the Fujinuma dam during the Tohoku (2011) strong ground motion. They used an old version of the Plaxis FE code and a Mohr-Coulomb soil model, with the addition of Rayleigh damping. The same model is used by previous research [65], which alleges a good consistence with the measured outcomes. The Plaxis dynamic module [50] utilizes the *hardening soil model by minor-strain stiffness* (hereinafter HS-small). The material method considers rather minor-strain stiffness and its non-linear appendance with a strain amplitude. The HS-minor is dependent on the nonlinear material model, in addition to two extra variants:

1. The first or rather minor-strain shear modulus, $G_{max}$
2. The secant shear modulus, $G$, is decreased to approximately 70% of $G_{max}$:

$$G = \frac{G_{max}}{1 + a\left|\frac{\gamma}{\gamma_{0.7}}\right|} \tag{9}$$

In which $\gamma$ is the average shear strain. Using $a = 0.385$ and $\gamma = \gamma_{0.7}$, $G \approx 0.722\, G_{\max}$ is collected. The HS-small model displays hysteresis in cyclic loading. Once performed in dynamic computations, the hysteric action of the HS-small model causes damping. The seismic behaviour of a numerical model is dependent on the regulation of a number of variants influencing the resources of energy loss in time-history analyses. The rate of damping displayed by a numerical model is maintained by the material damping, the integration plan of the formulas and the borderline cases [63].

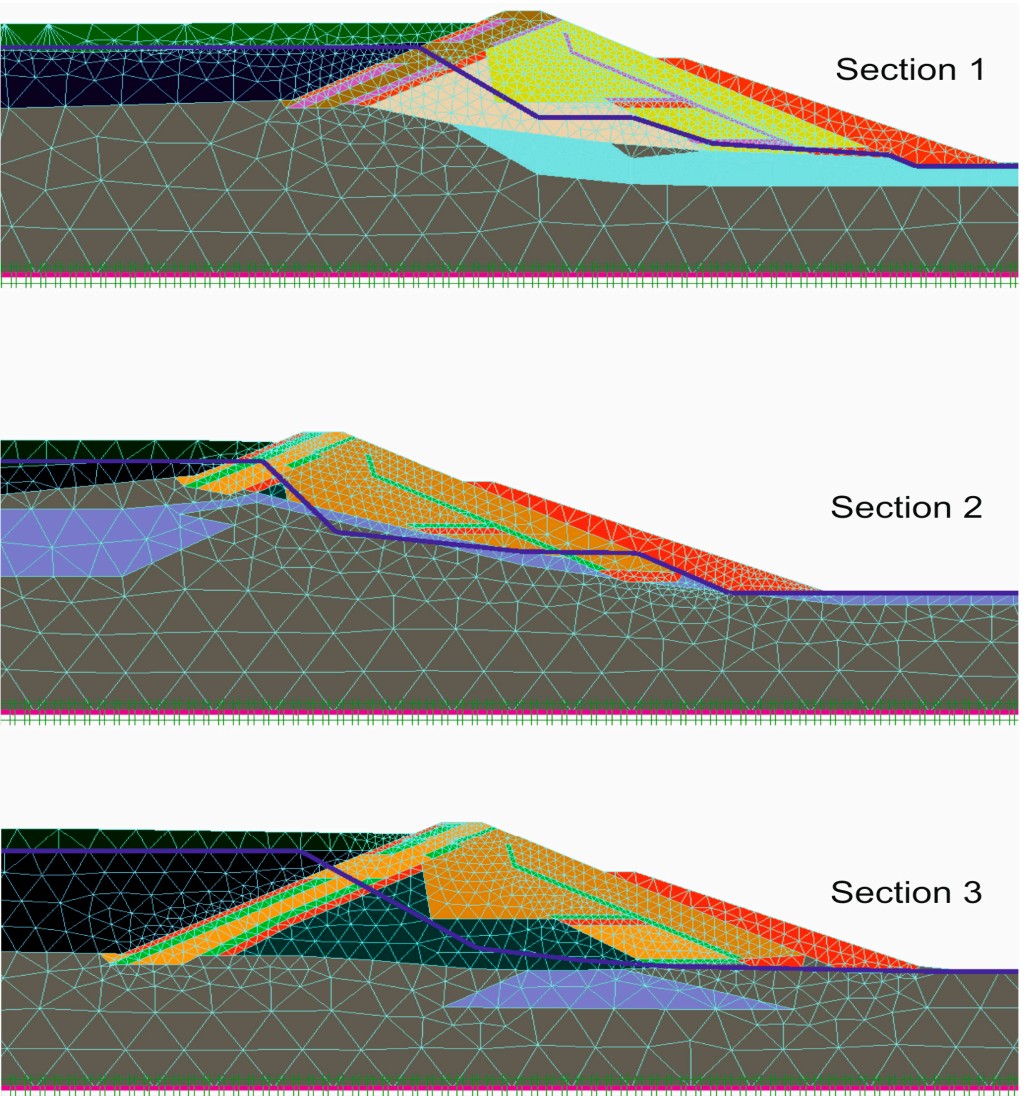

**Figure 6.** FE model for the three sections.

### 5.2. Finite Element Model

In this study, 15-noded triangular elements have been utilized due to their higher accuracy. These kinds of elements have 12 Gauss stress point. Table 6 shows the verification of the $V_s$, $f_r$, $\mathrm{AES}_{\min}$ and $\mathrm{AES}_{\mathrm{model}}$.

As far as the density of finite element refinement is concerned, the Average Element Size (AES) has been considered as the control parameter. According to previous research [66], the element dimension should not be greater than:

$$\lambda / 8 = V_s / (8 \cdot f_r) \tag{10}$$

where $\lambda$ is the wavelength with the highest frequency of relevance ($f_r$).

Table 6 shows the verification of the AES. Later it is explained how $V_s$ and $f_r$ have been calculated (see Section 5.2.2). Mesh coarseness has been chosen as fine, and it has been refined in the dam.

### 5.2.1. Material Models

So as to develop the dynamic calculation, two models were utilized: the Mohr-Coulomb (MC) and HS-Small. First, an MC model is created, due to the calculation time required by HS-Small. By means of the Mohr-Coulomb analysis, the available accelerograms have been applied to the three sections considered. Once the stresses have been examined, the critical accelerograms for each section have been selected. Then, the most critical cases have also been calculated by means of an HS-Small model. Table 7 shows some fundamental properties of the model.

**Table 7.** Fundamental characteristics of the hardening soil model.

| Characteristics | Related Parameters |
| --- | --- |
| Stress subjected to stiffness as regards a power law | $m$ |
| Plastic straining on account of first deviatoric loading | $E_{50}^{\text{ref}}$ |
| Plastic straining owing to first pressure | $E_{\text{oed}}^{\text{ref}}$ |
| Elastic reloading/unloading | $E_{\text{ur}}^{\text{ref}}, \nu_{\text{ur}}$ |
| Collapse as regards the MC failure criterion | $c$, $\varphi$ and $\psi$ |

Two additional variants:

$G_0$: First or rather minor-strain shear modulus.

$\gamma_{0.7}$: The secant shear modulus $G_s$ is decreased to approximately 70% of $G_0$. The original Hardening Soil model supposes elastic material behaviour in the course of reloading and unloading. On the other hand, the strain span where grounds could be regarded as completely elastic is quite little. With a rising strain amplitude, ground rigidness changes nonlinearly. It can also be noted that at the lowest strain which could be checked in a standard experiment, soil rigidness has frequently been reduced to less than half of its initial value. Hence, quite small-strain rigidity and its non-linear dependency on strain amplitude ought to be suitably considered. The HS-Small gives the likelihood of making it so. In addition, the HS-Small model displays hysteresis in cyclic loading. When it is applied in dynamic calculations, this hysteretic behavior leads to damping (based on the applied load and its equivalent strain amplitudes). All these facts make the HS-Small an appropriate model to be used in dynamic calculations, especially in order to obtain excess deformations.

### 5.2.2. Sources of Energy Dissipation

Material Damping

The hysteresis that the HS-Small model shows leads to an important material damping. Yet, under linear-elastic conditions (as in the Mohr-Coulomb model) hysteretic damping is zero. Nevertheless, many laboratory tests have demonstrated the existence of damping, even at much more minor strains [67,68]. In Plaxis, this problem is overcome by performing a visco-elastic material model, based on the Rayleigh equation:

$$C = \alpha_R M + \beta_R K \tag{11}$$

where:

$K$: Stiffness matrix

$M$: Mass matrix.

$C$: Damping matrix.

$\alpha_R$, $\beta_R$: Rayleigh coefficients. For a ground plate with a permanent rate of the damping parameter (*D*), the linear technique of formulations that gives the Rayleigh coefficients is:

$$\alpha_R + \beta_R \omega_{ni}^2 = 2 \cdot \omega_{ni} D \tag{12}$$

where:

*D* : Constant damping ratio.

$\omega$: Circular natural frequencies of the lamina:

$$\omega_{ni} = 2\pi f_{ni} \tag{13}$$

A damping ratio of 2% has been used according to previous research [63], who took this value as the damping ratio for a visco-elastic homogeneous plate lying on stiff base rock. Regarding the periods and natural frequencies, they could be computed as:

$$f_{ni} = \frac{V_{si}}{4h_i}(2n - 1) \tag{14}$$

*n*: the order number of the computed natural frequency. To overcome the fact of having several layers, a unique corresponding layer over stiff base rock has been utilised [69]. So, a uniform shear wave velocity has been calculated as a weighted average:

$$V_{sm} = \frac{1}{\sum_{i=1}^{N} h_i} \sum_{i=1}^{N} h_i V_{si} \tag{15}$$

Therefore, the natural period (*m* = 1) of the corresponding soil lamina is computed as:

$$T = 4 \sum_{i=1}^{N} \frac{h_i}{V_{si}} \tag{16}$$

Table 8 shows the different equivalent layers for each part.

**Table 8.** Equivalent layers for sections 1–3.

| Sections | Section 1 | | | Section 2 | | | Section 3 | | |
|---|---|---|---|---|---|---|---|---|---|
| Soil | *H* (m) | $\gamma$ (kN/m³) | $V_s$ (m/s) | *H* (m) | $\gamma$ (kN/m³) | *Vs* (m/s) | *H* (m) | $\gamma$ (kN/m³) | *Vs* (m/s) |
| Core | 3.30 | 19.80 | 95.74 | 3.9 | 19.8 | 95.74 | 3.9 | 19.8 | 95.74 |
| Filter | 1.10 | 20.00 | 95.26 | 1.1 | 20 | 95.26 | 2.06 | 20 | 95.26 |
| Quarry-run | 18.90 | 20.20 | 73.42 | 13.7 | 20.2 | 73.42 | 17.32 | 20.2 | 73.42 |
| Weathered Rock | 9.05 | 20.50 | 225.60 | 3.2 | 20.5 | 225.6 | 1.89 | 20.5 | 225.6 |
| Rockfill | 7.95 | 21.90 | 99.72 | - | - | - | 11.56 | 21.9 | 99.72 |
| Rock | 27.47 | 21.40 | 4118 | 46 | 21.4 | 4118 | 26.56 | 21.4 | 4118 |
| Equivalent layer | 67.77 | 20.90 | 1737.71 | 67.90 | 21.00 | 2822.30 | 63.29 | 20.99 | 1782.18 |

Finally, by means of Equations (8)–(12), input parameters $\alpha_R$ and $\beta_R$ are calculated (Table 9).

**Table 9.** Rayleigh coefficients.

| Rayleigh Coefficients | Section 1 | Section 2 | Section 3 |
|---|---|---|---|
| $\alpha_R$ | 1.20831 | 1.95873 | 1.32696 |
| $\beta_R$ | 0.00025 | 0.00015 | 0.00023 |

The amplification function for a soil plate on stiff base rock [70] allows calculation of the amplification factor for each frequency:

$$A(\mathrm{f}) = \frac{1}{\sqrt{\cos^2\left(2\pi\frac{h}{V_s}f\right) + \cos\left(2\pi\frac{hD}{V_s}f\right)^2}} \qquad (17)$$

The fundamental period varies depending on the section. While Sections 1 and 3 (rather similar) have almost the same fundamental period ($\approx$6 Hz); Section 2, which has a larger rock portion, has a higher one (11 Hz). This fact may be very important, because the most critical accelerograms can be different for each section.

Numerical Damping

The equation of the time integration is a significant element of the calculation process in the numerical application of dynamic issues. It strongly affects the steadiness and correctness of the operation. The Plaxis code implements the Newmark model implicit duration integration plan [71]. A reduced time step of $1/200$ s was used. In that technique, velocity and displacement of any point at $t + \Delta t$ are stated as:

$$u^{t+\Delta t} = u^t + \dot{u}^t \Delta t + \left[\left(\frac{1}{2} - \alpha_N\right)\ddot{u}^t + \alpha_N \ddot{u}^{t+\Delta t}\right]\Delta t^2 \qquad (18)$$

$$\dot{u}^{t+\Delta t} = \dot{u}^t + \left[(1 - \beta_N)\ddot{u}^t + \beta_N \ddot{u}^{t+\Delta t}\right]\Delta t \qquad (19)$$

The parameters that check the rectitude of the integration are $\alpha_N$ and $\beta_N$, the Newmark coefficients. Ref. [72] expressed these coefficients introducing a new parameter $\gamma$ (that could range from 0 to $1/3$):

$$\alpha_N = \frac{(1+\gamma)^2}{4} \qquad (20)$$

$$\beta_N = \frac{1}{2} + \gamma \qquad (21)$$

There are two general cases:

$\gamma = 0$. The modified method corresponds to the original Newmark method with a permanent average acceleration. No numerical damping is introduced.

$\gamma > 0$. The efficiency of the calculation improves. However, this method introduces numerical damping into the model. In order to avoid introducing numerical damping, the first option has been taken, that is:

$$\alpha_N = 0.25 \qquad (22)$$

$$\beta_N = 0.5 \qquad (23)$$

It is important to note that, while $\alpha_R$ and $\beta_R$ are material inputs, $\alpha_N$ and $\beta_N$ are calculation inputs.

Another important parameter in the accuracy control is the integration time step. The following formula [71] provides the crucial time step in a dynamic computation for one finite element:

$$dt_{\mathrm{crit}} = \frac{B}{\alpha\sqrt{V_P}\sqrt{1 + \frac{B^4}{4S^2} - \frac{B^2}{2S}\left(1 + \frac{1-2\nu}{4}\frac{2S}{B^2}\right)}} \qquad (24)$$

where:

$\alpha$ depends on the sort of element t.

For 15-nodes element $\alpha = 1/\left(19\sqrt{c_{15}}\right)$ being $c_{15} \approx 4.9479$ [73].

$V_P$ and $\nu$ rely on the type of soil, the pressure wave velocity and the Poisson ratio, respectively.

$B$ and $S$ are the surface of the element and the moderate length.

For an FE model, the crucial duration is the minimum of the individual critical times. For every FE model, Plaxis calculates the $dt_{\text{crit}}$ and sets the parameter control dynamic substeps to respect it. The rectitude of the computation and calculation time is strongly affected by this parameter. In this study, when the calculation time was too high, assuming $dt = dt_{\text{crit}}$, the time step was raised. Nevertheless, the accuracy of the calculation has always been checked.

The input accelerograms are introduced through ASCII or SMF format files. A displacement function of time is implied in the base.

Boundary Cases

An FE model requires boundary conditions able to simulate the energy absorption by the wave propagation in the base beyond the model's borders. In this way, Plaxis provides viscous absorbent boundaries, based on the reported [74] method. In this condition, ordinary and tangential stress components are described as follows:

$$\sigma_n = -c_1 \rho V_p \dot{u}_x \tag{25}$$

$$\tau = -c_2 \rho V_s \dot{u}_y \tag{26}$$

$V_p$, $V_s$ and $\rho$ are material properties: compressions and shear waves, and density. Standard settings have been taken in order to choose the boundary parameters $c_1$ and $c_2$ (relaxation coefficients). These values, proposed by previous research [50], are:

$$c_1 = 1 \tag{27}$$

$$c_2 = 0.25 \tag{28}$$

Yet, it is unlikely to express that shear waves are totally absorbed, and there are not clear criteria to determine $c_1$ and $c_2$. In order to overcome these uncertainties, Ref. [75] suggested moving the borders adequately far away from the related region. In fact, the dam model has been widened up to thrice the original width.

*5.3. Calculation Steps*

As stated above, the calculation has been carried out under undrained conditions. Being a finished structure, no construction stages have been considered, but rather the finished dam. In the three Mohr-Coulomb models, all the available accelerograms have been applied. Nonetheless, the HS-Small model has only been calculated with the most critical accelerograms. The calculation phases are:

(a)  Self-weight.
(b)  Self-weight and accelerograms application.

**6. Calculation Results**

Once the critical accelerograms have been selected and applied, the results have been studied. First, simulations of relative shear stress and plastic points have been made and used to find the critical moment of the earthquake for each section. Then, a thorough study of these cases has been carried out. After, an analysis of stress has been implemented, followed by plastic points examinations. Special attention has been paid to the appearance of a slip or failure surface. Finally, deformations and total displacements have been observed and interpreted.

*6.1. Analysis of Stress*

So as to analyze the static balance of the dam, the relative shear stress has been used. This is defined as follows:

$$\tau_{\text{rel}} = \frac{\tau}{\tau_{\text{max}}} \tag{29}$$

where:

$\tau$: Shear stress.

$\tau_{\max}$ : Shear strength.

Figure 7 displays the critical case for the three sections. Examining this figure, it can be stated that the bedrock is resistant enough to overcome the earthquake. Thereby, the bigger the bedrock zone in the dam body, the lesser the failure risk. So, comparing relative shear stress in Section 2, which has a large proportion of bedrock, to $\tau$ in the other sections, it is clear that this is not the critical section. Although the higher relative shear stresses appear in Section 1, no failure surface seems to cross the dam body. Nor does a slip surface appear in the other two cases. It can be observed that the *Las Viñas* fill may exceed its resistance, which does not imply the instability of the dam but just that of this material situated upstream.

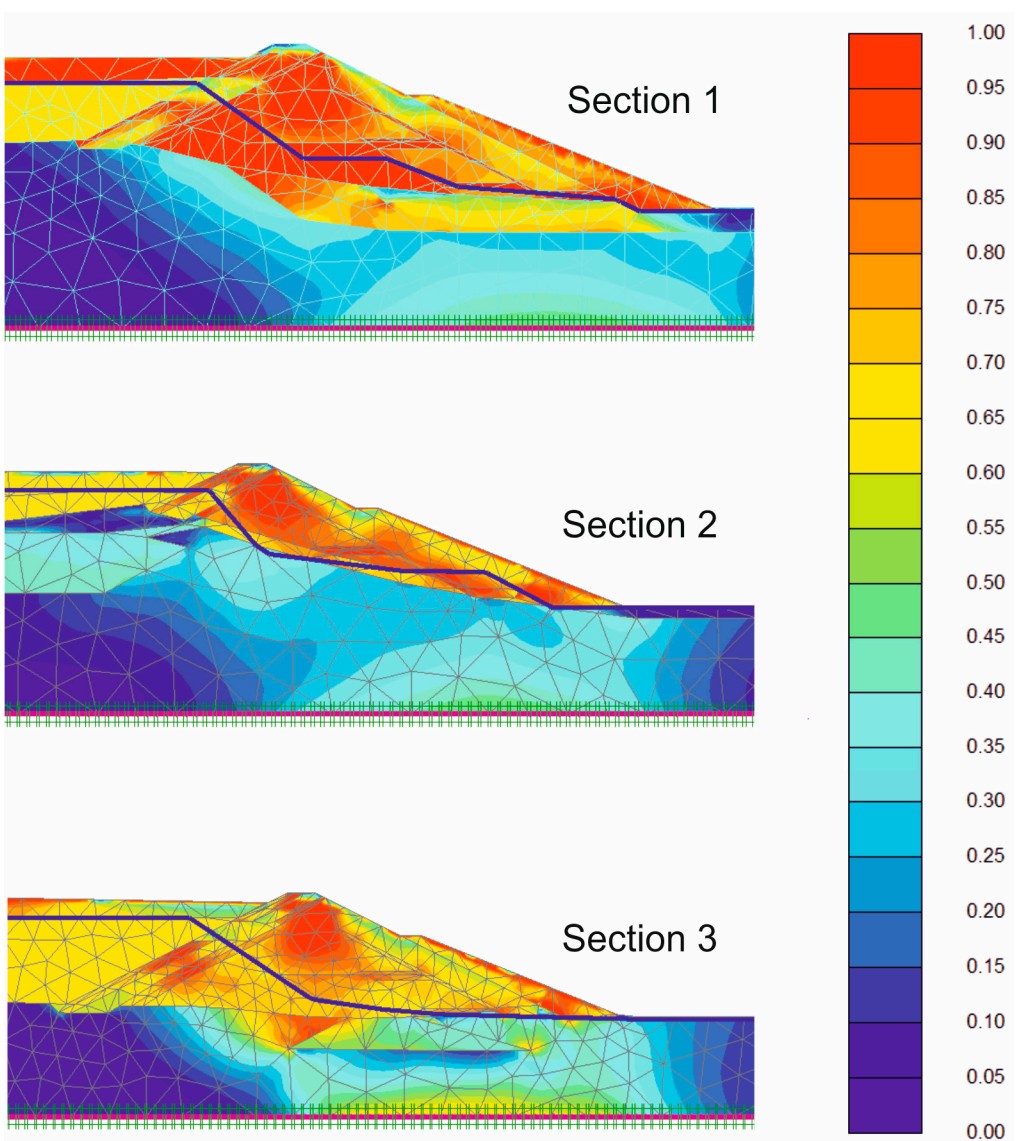

**Figure 7.** Relative shear stress for the three sections at the most unfavorable instant with the HS-Small model.

While the largest shear stress occurred in section 1, it was obtained in section 2 and then section 3, respectively. In order to check the dam resistance, MC and cut-off places have been also analyzed (Figure 8). It should be noted that the plastified zones are scattered and do not produce a continuous sliding surface. This failure situation only lasts for a few moments, which is not a true failure situation. Therefore, a dam failure is not produced.

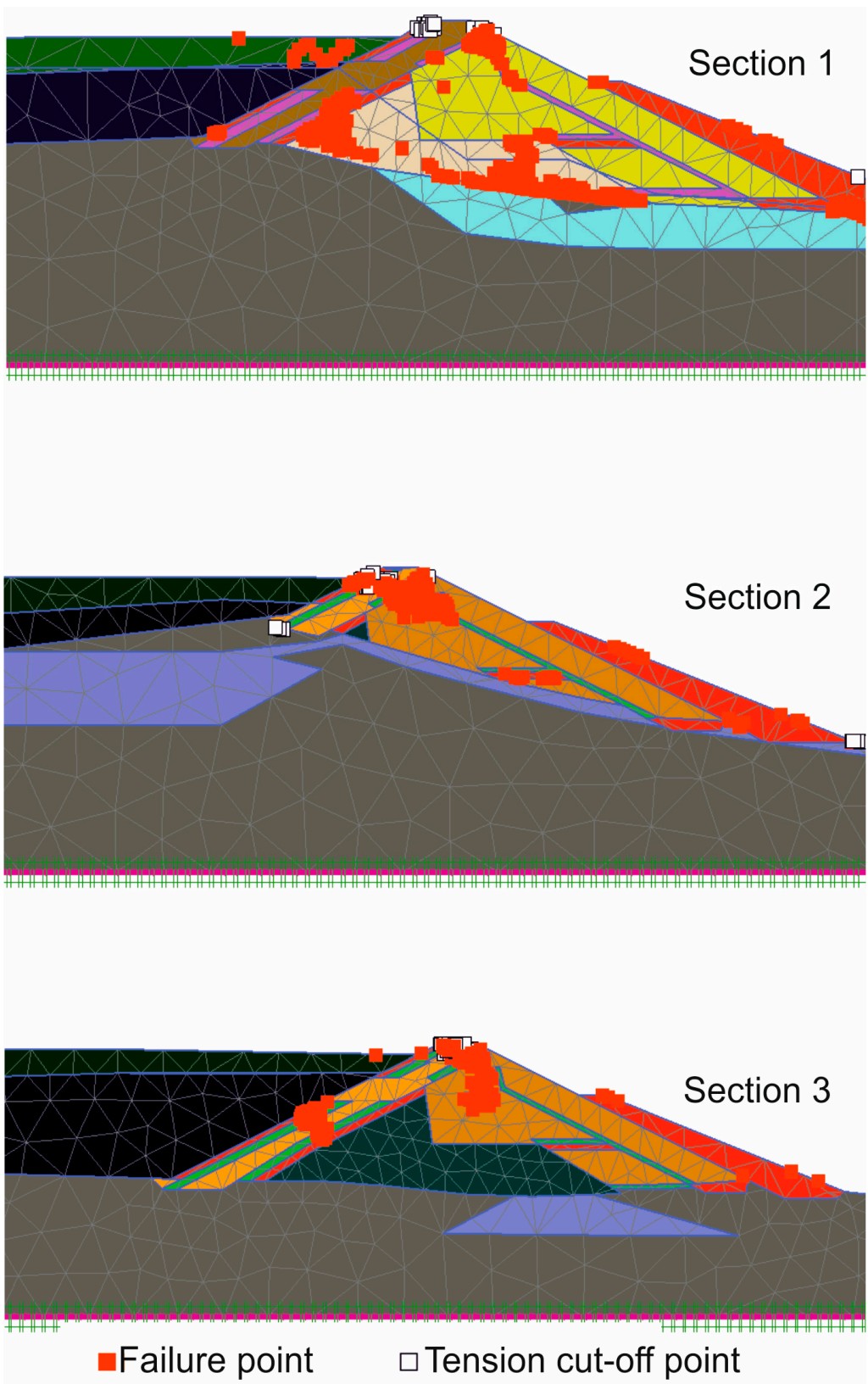

**Figure 8.** Tension cut-off points and MC points for the three sections at the most unfavorable instant with the HS-Small model.

A point is denoted as a Mohr-Coulomb point if, due to its stress state, it is currently on the MC envelope. Meanwhile, tension cut-off places are those which have reached the tension cut-off criterion. In this situation, the allowed tension is equal to zero. As has been

proved by means of the shear relative stress outputs, it can be noted that no slip surface appears. Nevertheless, the dynamic load is not negligible at all. For example, Section 1 shows several plastic zones at the same time. If two or more zones come together, a failure surface may appear, or the dam could even collapse.

### 6.2. Analysis of Deformations

A variety of simulations have been made to find out the displacement over time. Figure 9 shows the deformed meshes. It can be observed that vertical displacements are negligible in the dam body and that it remains almost unaltered.

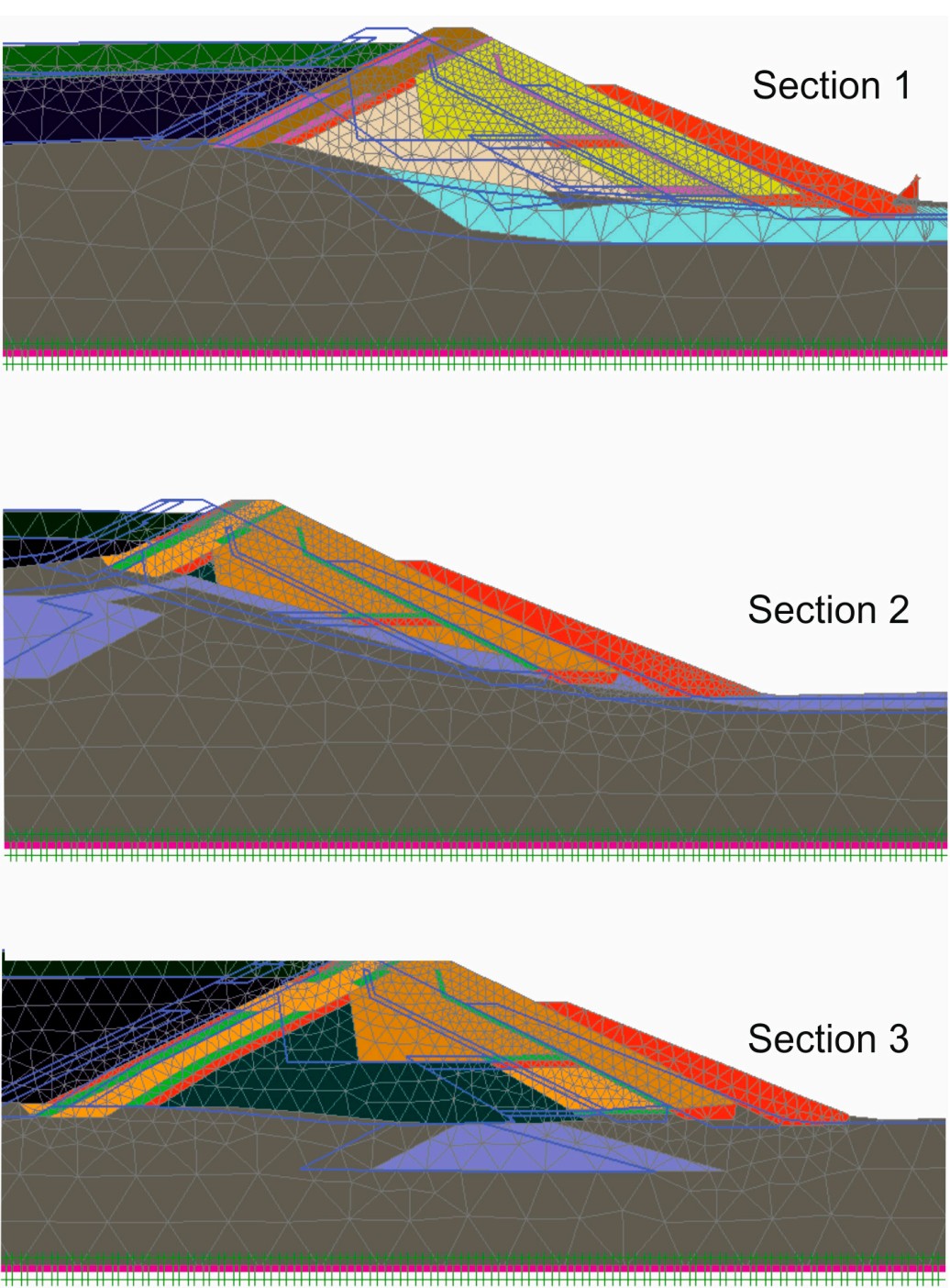

**Figure 9.** Deformed mesh for the three sections.

The residual deformation at a point can be obtained as the difference between its displacement and the displacement at the base (which is fixed). The HS-small model gives reliable values of these deformations. The residual deformation values, due to the action of the accelerograms, obtained between a point placed on the top of the dam and the base are quite small: 2 mm in Section 1; 3 mm in Section 2; and 2.5 mm in Section 3. It can also be observed that the *Las Viñas* fill and the tailings suffer a large deformation which was expected due to its low resistance.

## 7. Analysis of Liquefaction

Soil liquefaction can cause significant damage when an earthquake takes place. This physical phenomenon, which occurs during some earthquakes, leads the soil to a temporary loss of strength. Consequently, loose cohesionless saturated sandy (even silty) soils behaves like a viscous liquid. Pore pressure may increase when shear waves pass through a saturated sandy soil. This can make the effective stress drop to zero and consequently eliminate the shear strength. If the pore pressure has increased enough to decrease the shear strength to zero, the soil is no longer able to support whatever is above (overlying soil layers, buildings, etc.) and it flows like a liquid. According to Eurocode 8, part 5, article 4.1.4 (2):

> "*An evaluation of the liquefaction susceptibility shall be made when the foundation soils include extended layers or thick lenses of loose sand, with or without silt/clay fines, beneath the water table level, and when the water table level is close to the ground surface. This evaluation shall be performed for the free-field site conditions (ground surface elevation, water table elevation) prevailing during the lifetime of the structure*".

In this case, the foundation soils do not contain extended plates of thick lenses of loose sand at all (see Section 2). Instead, the dam layers over rigid bedrock and the materials inside the dam are quite compacted. However, this is not the only way of liquefaction that affects the Almagrera tailings dam. Although there are not materials susceptible to liquefying below or inside the dam, either tailings or the *Las Viñas* materials may fail by liquefaction. A priori, a failure inside the reservoir is not so important. Yet, if the fill behaves like a liquid it can greatly increase the pressure on the upstream dam side.

Santucci et al. [76] published a database for peak ground acceleration (PGA) threshold in a liquefaction event. The database included 201 liquefaction cases caused by earthquakes. In the database, 40% of the data were related to strong ground motions with moment magnitudes from 6.8 to 7.1, 13% were related to quite big incidents (magnitudes of 8 or over), and 2% were related to magnitudes ranging between 5.9 and 6.2. As mentioned in Section 4.2 a magnitude larger than 5.5 inside the South Portuguese unit, where Almagrera is located, is improbable. This paper provided a graph of the cumulative distribution of liquefaction occurrence which was empirical and fitted to a beta probability distribution (Figure 10). Finally, it fixes a threshold of PGA equal to 0.09 g at the site surface as the limit below which no liquefaction occurs.

Justo et al. [77] published a thorough study about these tailings and the *Las Viñas* material. In that study, the *Las Viñas* material was, on the one hand, classified between silty sand and sandy silt. On the other hand, the mine tailings are quite heterogeneous and may vary among sandy tailings without plasticity (ML), silt with a high liquid limit (MH) and silty clay (CL). Since there are no plastic zones and loose materials and some of them are under the water table, liquefaction could happen inside the old reservoir. Nevertheless, although some materials inside the reservoir are susceptible to liquefying, soil liquefaction would only occur under certain seismic conditions (magnitude, peak acceleration, earthquake duration, etc.). As stated before, the Almagrera dam is not located in a critical seismic zone.

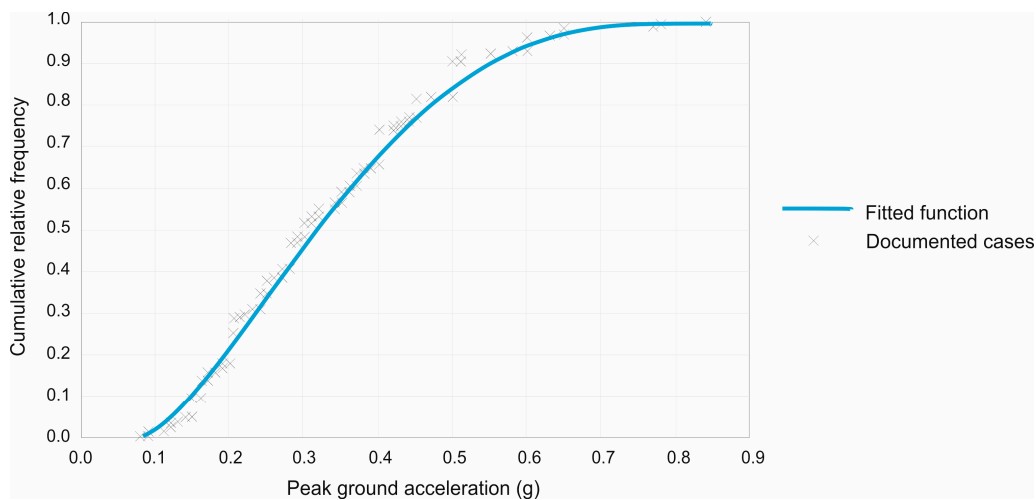

**Figure 10.** Accumulative distribution of liquefaction happening suited to a 4-parameter Beta possibility distribution [76].

Thereby, taking advantage of the dynamic calculations, the curves of tailings acceleration have been examined. Figure 11 provides the acceleration-against-time graph at one point on the old reservoir surface for the most critical situation (accelerogram 607 and Section 2). It can be observed that the threshold value of 0.09 g has not been exceeded at the site surface. The dynamic calculation carried out in Section 5 does not include a liquefaction analysis. However, as stated above in this section, this is not necessary for the Almagrera tailings dam. Regarding the material characteristics, it is very improbable for liquefaction to occur in the base soil or the dam body. With regard to the upstream materials, they could be susceptible to liquefaction. All the same, the seismicity at the site is quite minor. Evaluating the acceleration on the deposits surface, this is well below the established PGA thresholds. In addition, the tailings drainage will continue, which means that the water table should drop below the tailings, reducing the liquefaction possibility. In conclusion, after evaluating the Almagrera tailings dam and comparing it with some liquefaction cases, it can be stated that the risk of liquefaction is negligible.

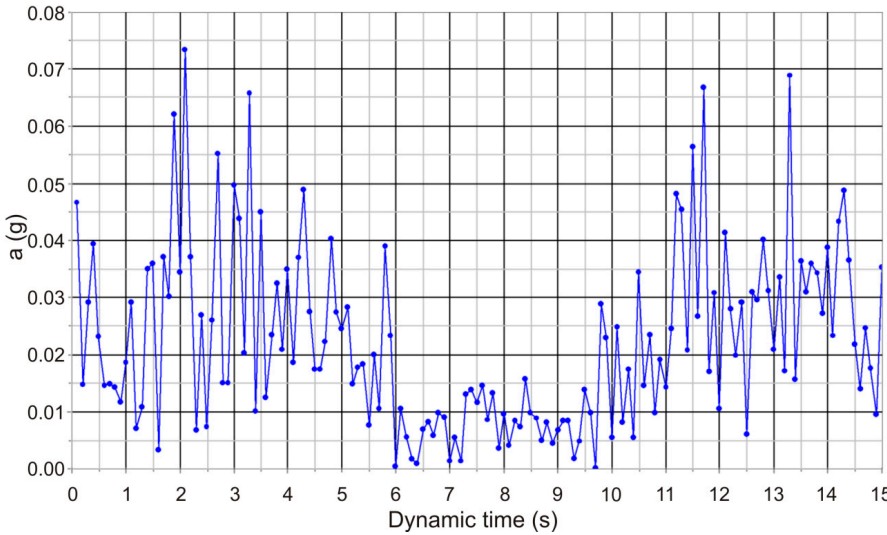

**Figure 11.** Acceleration vs. time for a point at the surface of the tailings for Section 2.

## 8. Conclusions

(1)  A dynamic computation for the dry closure of the Almagrera's tailings dam is presented herein.

(2) In order to carry out a dynamic seismic analysis, a method for selecting real accelerograms has been presented. The method depends upon the construction of USHARS for the Almagrera site, considering the kind of ground at the foundation and the required hazard level. Up to seven accelerograms from the European Strong Motion data base have been selected.

(3) The main advantage of using dynamic FE analysis is that it provides detailed information of the soil stress distribution and deformation. It should be noted that special care must be taken with the model. The material damping, the numerical damping and the boundary conditions have been configured to fit the dam's real conditions. The HS-small model used allows calculating the residual deformation. The results show that the dam is safe enough.

(4) The expected permanent displacement of the dam body after the maximum expected earthquake is acceptable, with Section 1 being the most critical. The results show that no continuous failure surface crosses the dam body for the maximum expected earthquake. Nonetheless, the results show that, at some short moments the dam body is close to collapse. Moreover, in some places, Las Viñas fill and the tailings may attain failure. This indicates a tailings displacement, but not a dam failure. The liquefaction that could suffer the fill upstream is unimportant.

**Author Contributions:** Conceptualisation, A.M.-E. and J.L.d.J.A.; methodology, A.M.-E.; validation, A.M.-E., J.L.d.J.A. and P.C.; formal analysis, A.M.-E.; investigation, A.M.-E. and P.C.; resources, J.L.d.J.A.; data curation, A.M.-E. and P.C.; writing—original draft preparation, A.M.-E. and P.C.; writing—review and editing, A.M.-E. and M.K.; visualisation, A.M.-E. and M.K.; supervision, A.M.-E.; project administration, J.L.d.J.A.; funding acquisition, J.L.d.J.A. All authors have read and agreed to the published version of the manuscript.

**Funding:** This research was funded by the Spanish Ministry of Science and Innovation through the project BIA2010-20377.

**Data Availability Statement:** Data available upon request to the authors.

**Conflicts of Interest:** The authors declare no conflicts of interest.

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
