# Peer review of "Dynamic Analysis of the Almagrera Tailings Dam with Dry Closure Condition"

_sustainability, doi:10.3390/su16041607_

Round 1

Reviewer 1 Report

Comments and Suggestions for Authors

This article investigates the dynamic behavior of the dry closure of the Almagrera tailings dam under an earthquake effect using the Plaxis 2D programme. some comments are listed below for improving the quality of this article. 

1.  The validation of the Plaxis 2D programme should be included in this paper. If there is some test evidence in other articles, it could be better to cite these article for the validation. 

2.  The Abstract section should be shorted for highlighting the novelty of the article. 

3.   Neither the tailings nor the dam is expected to suffer liquefaction. what is the engineering meaning of this result? 

4.  A copper tailings dam located in China was statically investigated [5]. [6] examined the impact of seepage on the structure of a mine waste fill dam. 

How to understand this sentence ? [6] examined ??? 

5. FIgure 3 can be re-print for the better reading effect. 0.5-3.0 for Y axis and 0.04-0.3 for X axis labels. 

6. 15-noded triangular elements are applied in this paper. and I think it would be better to use the Quadrilateral or Hexahedron elements insted. Please considere this comment. 

7.  It can be seen from figure 6 that the failure points are dispersed in the figure. why? And how to judge the failure point and what is the failure criterion? 

8. Conclusion section is too long and for the better reading effect, some sentences are suggested to be lised in This section by numbering (1)  (2)   (3)...

Comments on the Quality of English Language

IT can be improved and rewritten by a native English speaker. 

Author Response

Reviewer 1:

Comments and Suggestions for Authors

This article investigates the dynamic behavior of the dry closure of the Almagrera tailings dam under an earthquake effect using the Plaxis 2D programme. some comments are listed below for improving the quality of this article. 

We would like to thank the reviewer for his/her comments. We think that the manuscript has improved. Please, see next the corrections/modifications done accordingly.

  1. The validation of the Plaxis 2D programme should be included in this paper. If there is some test evidence in other articles, it could be better to cite these article for the validation. 

The following text has been included in the introduction Section of the manuscript:

“An excellent fit is achieved with displacements and a reasonable one with pore water pressures in a dam using Plaxis 2D [37].

Reference: Da Silva, E.M., Justo, J.L, Durand, P., Justo, E., Vazquez-Boza, M. (2017). The effect of geotextile reinforcement and prefabricated vertical drains on the stability and settlement of embankments. Geotextiles and Geomembranes, 447-461.

  1. The Abstract section should be shorted for highlighting the novelty of the article.

The abstract has been shortened for highlighting the novelty of the article to:

“The Spanish authorities have become more sensitive to the problem of ore tailings following the Aznalcollar dam break. The main novelty of this paper is to study the dynamic behavior for the dry closure of the Almagrera dam under the action of an earthquake. This study has been carried out with the Plaxis 2D program, which uses the finite element (FE) method. The dynamic analysis of the dam has been interpreted in terms of deformations, displacements and principal stresses. Also, the construction of the Uniform Seismic Hazard Acceleration Response Spectrum (USH-ARS) and the selection of real accelerograms for the time-history dynamic calculations is a noted novelty. Numerical analyses show that the dam is safe enough, because a failure surface has not been formed, although several plastic zones may appear in the dam. The FE study of deformations displays that the tailings may attain large deformations, displacements and failure, although this does not jeopardize the safety of the dam, where the displacements are smaller than 3 mm. Neither the tailings nor the dam is expected to suffer liquefaction. It is determined that the 0.09 g threshold value is not exceeded in the acceleration-time graphs on the old reservoir field surface, which is the most critical situation.”

Also, the main novelties of this manuscript have been described in detail in the last paragraph of the Introduction.

  1.  Neither the tailings nor the dam is expected to suffer liquefaction. what is the engineering meaning of this result? 

It means that the liquefaction can lead to failure. A comment has been included in abstract to clarify this issue: “It is determined that the 0.09 g threshold value is not exceeded in the acceleration-time graphs on the old reservoir field surface, which is the most critical situation.”

  1. A copper tailings dam located in China was statically investigated [5]. [6] examined the impact of seepage on the structure of a mine waste fill dam. 

How to understand this sentence ? [6] examined ??? 

The text has been corrected to:

“A copper tailings dam in China was statically investigated [5]. Yin et al. [6] studied the impact of seepage on the structure of a mine waste dam.”

  1. FIgure 3 can be re-print for the better reading effect. 0.5-3.0 for Y axis and 0.04-0.3 for X axis labels. 

The reviewer is right and the figure has been modified accordingly. Also, different types of lines have been used, as suggested by Reviewer#2.

New Figure 3 (now 5).

  1. 15-noded triangular elements are applied in this paper. and I think it would be better to use the Quadrilateral or Hexahedron elements insted. Please considere this comment. 

The reviewer is right as hexahedron or quadrilateral elements would improve the computer burden required. However, 15-noded triangular elements have been used due to their high accuracy (Please, see Section 3.5 and Section 5.2).

  1. It can be seen from figure 6 that the failure points are dispersed in the figure. why? And how to judge the failure point and what is the failure criterion? 

Precisely because the plastified zones are scattered and do not produce a continuous sliding surface, there is no dam failure. The failure points are Mohr-Coulomb points. In a dynamic analysis, the best failure criterion must be based upon displacements, because as Ambraseys already said, a failure situation that only lasts for a few instants is not a true failure situation. A comment to clarify this issue has been added in the text:

“It should be noted that the plastified zones are scattered and do not produce a contin-uous sliding surface. The failure situation only lasts for a few instants, which is not a true failure situation. Therefore, a dam failure is not produced.”

  1. Conclusion section is too long and for the better reading effect, some sentences are suggested to be lised in This section by numbering (1)  (2)   (3)...

The text has been modified according to the reviewer´s suggestion to:

1) A dynamic computation for the dry closure of the Almagrera’s tailings dam is pre-sented herein.

2) In order to carry out a dynamic seismic analysis, a method for selecting real acceler-ograms has been presented. The method depends upon the construction of USHARS for the Almagrera site, considering the kind of ground at the foundation and the re-quired hazard level. Up to seven accelerograms from the European Strong Motion data base have been selected.

3) The main advantage of using dynamic FE analysis is that it provides detailed infor-mation of the soil stress distribution and deformation. It should be noted that special care must be taken with the model. The material damping, the numerical damping and the boundary conditions have been configured to fit the dam’s real conditions. The HS-small model used allows calculating the residual deformation. The results show that the dam is safe enough.

4) The expected permanent displacement of the dam body after the maximum ex-pected earthquake is acceptable, with Section 1 being the most critical. The results show that no continuous failure surface crosses the dam body for the maximum ex-pected earthquake. Nonetheless, the results show that, at some short moments the dam body is close to collapse. Moreover, in some places Las Viñas fill and the tailings may attain failure. This indicates a tailings displacement, but not a dam failure. The lique-faction that could suffer the fill upstream is unimportant. 

Reviewer 2 Report

Comments and Suggestions for Authors

Dear Authors, I think it's good work. However, the following notes and modifications are suggested:

1. You need to refine your abstract to make it more interesting to the reader, highlight your novelty, and provide significantly more quantitative information.

2. The introduction does not show that:

The paper is an original contribution to the field, and the information presented is new. I.e., what is lacking in previous work? "Novelty statement"

3. I think that the method of placing references is not appropriate, for example ([15]; [16]; [17]; [18]; [19]; [20]; [21]; [22]; [23]; [24]; [25]; [26]; [27]; [28]).

it should be [15–28],.......................etc.

4. The authors should be consistent when introducing formulas. All formulas should have references. If they are derived, the derivation should be presented. All variables in all formulas need to be described in units.

5. The language has grammatical errors. The wrong tense is used. The future tense does not belong in a scientific article except in the conclusions. Some examples:

a. "accumulated in its tailings will be extracted and that it will be waterproofed to avoid water” in L149, p.4. and “In all events, that category will contain entire” in L174, p.5................etc.

6. All Figures should be improved, made bigger, and made clearer, i.e. high resolution should be used.

7. In Figure 3, you must change the types of lines used in drawing, not just change the colors. For example, you can use dashed, continuous, and dotted lines and use different symbols.

8- It is best to make the grid inside the figure.3 and figure.9 in the form of dashed, low-density lines to see the studied lines better, reducing empty spaces inside the figure.3 and increasing axis data.

9- You should put the legend for the figure.8. 

10- Show the mesh in one figure, not for all figures 4, 5, 6, and 7.

11- Mesh construction plays a vital role in the quality of the resulting simulation data. It is necessary to present mesh images showing the stages of mesh quality from coarse to fine. What is the mesh quality? Simulation becomes essential concerning mesh construction near the studied area.

12- Mesh independence has not been studied. A figure must be provided.

13- You must refer to the table before mentioning it in the text, for example "Table 6 shows the verification of the AES"

14- Did you use a small or large time step size for your solution? How was the stability? Is the system stiff or not?

15- The conclusions must be more specific about the obtained results. The present conclusions are more general. This section must be revised, and authors need to state their key findings and are the conclusions consistent with the evidence and arguments presented..

16- Maybe there is some problem with the size of lines from 455 to 465.

In my opinion, this work can be published after major corrections. Hence, I recommend Major revision of this manuscript. The authors should revise the manuscript as per the above suggestions and improve the write up of the entire manuscript.

Comments on the Quality of English Language

Average quality.

Author Response

Reviewer 2:

Comments and Suggestions for Authors

Dear Authors, I think it's good work. However, the following notes and modifications are suggested:

We would like to thank the reviewer for his/her comments and cheers. We think that the manuscript has improved. Please, see next the corrections/modifications done accordingly.

  1. You need to refine your abstract to make it more interesting to the reader, highlight your novelty, and provide significantly more quantitative information.

The abstract has been shortened and revised. Also, more quantitative results have been included, emphasizing its originality:

“The Spanish authorities have become more sensitive to the problem of ore tailings following the Aznalcollar dam break. The main novelty of this paper is to study the dynamic behavior for the dry closure of the Almagrera dam under the action of an earthquake. This study has been carried out with the Plaxis 2D program, which uses the finite element (FE) method. The dynamic analysis of the dam has been interpreted in terms of deformations, displacements and principal stresses. Also, the construction of the Uniform Seismic Hazard Acceleration Response Spectrum (USH-ARS) and the selection of real accelerograms for the time-history dynamic calculations is a noted novelty. Numerical analyses show that the dam is safe enough, because a failure surface has not been formed, although several plastic zones may appear in the dam. The FE study of deformations displays that the tailings may attain large deformations, displacements and failure, although this does not jeopardize the safety of the dam, where the displacements are smaller than 3 mm. Nei-ther the tailings nor the dam is expected to suffer liquefaction. It is determined that the 0.09 g threshold value is not exceeded in the acceleration-time graphs on the old reservoir field surface, which is the most critical situation.”

Also, the main novelties of this manuscript have been described in detail in the last paragraph of the Introduction.

  1. The introduction does not show that:

The paper is an original contribution to the field, and the information presented is new. I.e., what is lacking in previous work? "Novelty statement"

The main novelties of this manuscript have been described in detail in the last paragraph of the Introduction and in the new abstract.

  1. I think that the method of placing references is not appropriate, for example ([15]; [16]; [17]; [18]; [19]; [20]; [21]; [22]; [23]; [24]; [25]; [26]; [27]; [28]).

it should be [15–28],.......................etc.

Multiple reference notation has been corrected to [15-28].

  1. The authors should be consistent when introducing formulas. All formulas should have references. If they are derived, the derivation should be presented. All variables in all formulas need to be described in units.

The references of all formulas have been checked and verified, and the units of all variables in all formulas are presented.

Reference number 49 has been added to the manuscript for formulas.

Amouzou, G.Y.; Soulaïmani, A. Numerical Algorithms for Elastoplacity: Finite Elements Code Development and Implementation of the Mohr–Coulomb Law. Appl. Sci. 2021, 11, 1–27, doi:10.3390/app11104637.

The exponent expressions in γ (kN/m3) units in Table 8 have been corrected.

  1. The language has grammatical errors. The wrong tense is used. The future tense does not belong in a scientific article except in the conclusions. Some examples:
  2. "accumulated in its tailings will be extracted and that it will be waterproofed to avoid water” in L149, p.4. and “In all events, that category will contain entire” in L174, p.5................etc.

The manuscript was checked for future tense. The specified corrections were made and added to the manuscript.

The dry closure of a dam means that all the water accumulated in its tailings is going to be extracted and that it is going to be waterproofed to avoid water entering in the future.

In all events, that Category contains entire rafts and dams not counted in categories A or B.

The whole manuscript has been reviewed by a native English professional reviewer, who has made some corrections.

  1. All Figures should be improved, made bigger, and made clearer, i.e. high resolution should be used.

All figures have been upgraded as much as possible, as suggested. They have been enlarged and made more understandable and high resolution has been used. All figures have been redone from the original sources and made at 300 dpi in TIFF format.

  1. In Figure 3, you must change the types of lines used in drawing, not just change the colors. For example, you can use dashed, continuous, and dotted lines and use different symbols.

Figure 3 (now 5) has been revised according to the reviewer´s recommendations.

New Figure 3 (now 5).

  1. It is best to make the grid inside the figure.3 and figure.9 in the form of dashed, low-density lines to see the studied lines better, reducing empty spaces inside the figure.3 and increasing axis data.

Figure 3 (now 5) and Figure 9 (now 11) have been modified according to the reviewer´s suggestion. The grid-lines have been reduced to low density grey lines.

New Figure 9 (now 11).

  1. You should put the legend for the figure.8. 

Figure 8 (now 10) has been completely depicted at 300 dpi and a graphical legend has been added.

New Figure 8 (now 10).

  1. Show the mesh in one figure, not for all figures 4, 5, 6, and 7.

Sorry, but the authors don´t agree with this comment. The mesh size is a key point in Finite Element calculation (Section 5.2 Finite element model) and it is an information that must be clearly studied and shown. The common in FE manuscripts is to show the mesh. Also, that would make the three sections to look different, which could be misleading.

  1. Mesh construction plays a vital role in the quality of the resulting simulation data. It is necessary to present mesh images showing the stages of mesh quality from coarse to fine. What is the mesh quality? Simulation becomes essential concerning mesh construction near the studied area.

It should be noted that a non-local algorithm is implemented into Plaxis, so that the softening process can be analysed without mesh dependency. The mesh is automatically generated in Plaxis. The mesh coarseness has been chosen as fine in this research and it has been refined in the dam. To check its validity, the Average Element Size (AES) has been calculated. To do so, the minimum AES has been verified to be minor than that of the model (please, see Table 6 in Section 5.2).

  1. Mesh independence has not been studied. A figure must be provided.

Please, see Query#11.

  1. You must refer to the table before mentioning it in the text, for example "Table 6 shows the verification of the AES"

The reviewer is right. The following text has been added: “Table 6 shows the verification of the AESmin and AESmodel.”

  1. Did you use a small or large time step size for your solution? How was the stability? Is the system stiff or not?

A reduced time step of 1/200 seconds was used.

The formulation of the integration with time constitutes an important factor in the stability and accuracy of the calculation process. Implicit and explicit methods can be used. The advantage of the explicit integration is that is relatively simple to formulate. However, the disadvantages are that the calculation process is not robust and several limitations are imposed in the calculation steps. Implicit methods are more complicated, but produce a more reliable calculation process and normally more accurate. In this study, Newmark's implicit time integration system has been used. With this method, the displacement and the velocity at the point in time t+Δt are expressed, respectively as stated in Equations 18-19.

             (18)

                                          (19)

In the text, it is also stated:

Another important parameter in the accuracy control is the integration time step. The following formula [72] gives the crucial time step in a dynamic computation for one finite element:

                          (24)

where:

 depends on the sort of element t.

For 15-nodes element  being  ([73]).

 and  rely on the type of soil, the pressure wave velocity and the Poisson ratio, respectively.

 and   are the surface of the element and the moderate length.

For an FE model, the crucial duration is the minimum of the individual critical times. For every FE model, Plaxis calculates the  and sets the parameter control dynamic substeps to respect it. The rectitude of the computation and calculation time is strongly affected by this parameter. In this study, when the calculation time was too high, assuming , the time step was raised. Nevertheless, the accuracy of the calculation has been always checked.

The input accelerograms are introduced through ASCII or SMF format files. A displacement function of time is implied in the base.

  1. The conclusions must be more specific about the obtained results. The present conclusions are more general. This section must be revised, and authors need to state their key findings and are the conclusions consistent with the evidence and arguments presented.

The results regarding the results obtained are explained more specifically. This chapter has been revised and the authors' main findings are stated. It is interpreted whether the results are consistent with the evidence and arguments presented.

1) A dynamic computation for the dry closure of the Almagrera’s tailings dam is pre-sented herein.

2) In order to carry out a dynamic seismic analysis, a method for selecting real acceler-ograms has been presented. The method depends upon the construction of USHARS for the Almagrera site, considering the kind of ground at the foundation and the re-quired hazard level. Up to seven accelerograms from the European Strong Motion data base have been selected.

3) The main advantage of using dynamic FE analysis is that it provides detailed infor-mation of the soil stress distribution and deformation. It should be noted that special care must be taken with the model. The material damping, the numerical damping and the boundary conditions have been configured to fit the dam’s real conditions. The HS-small model used allows calculating the residual deformation. The results show that the dam is safe enough.

4) The expected permanent displacement of the dam body after the maximum ex-pected earthquake is acceptable, with Section 1 being the most critical. The results show that no continuous failure surface crosses the dam body for the maximum ex-pected earthquake. Nonetheless, the results show that, at some short moments the dam body is close to collapse. Moreover, in some places Las Viñas fill and the tailings may attain failure. This indicates a tailings displacement, but not a dam failure. The lique-faction that could suffer the fill upstream is unimportant.

16- Maybe there is some problem with the size of lines from 455 to 465.

Font sizes have been revised from 9 to 10.

Reviewer 3 Report

Comments and Suggestions for Authors

This paper presents the dynamic behavior of the dry closure of the Almagrera tailings dam under an earthquake effect. The authors have carried out the investigation via Plaxis 2D program, which uses the finite element method. Three different FE models of unfavorable sections have been studied for the dam body and the tailings. Though the subjected investigation is worthwhile, there are several deficiencies in the manuscript. A major revision of the manuscript is required.

1.     Beginning of the abstract need revision.

2.     It is recommended to incorporate the critical quantitative results of the study in the abstract.

3.     The novelty of the present study lacks clarity. The authors should explicitly establish, in alignment with the existing technical literature relevant to this paper, the innovative contributions of the article in comparison to other research efforts.

4.     It is advisable to avoid the compounding of references.

5.     The literature review provided in the introduction section lacks comprehensiveness. It is advisable to critically analyze previous studies in detail, rather than merely summarizing the actions of researchers in the past.

6.      It is recommended to add recent studies on the dynamic analyses of the rockfill dams using the finite element (FE) method. 

7.     My main concern regarding the manuscript is that the authors draw conclusions about the structural stability of a highly critical structure based solely on a 2D analysis. In my perspective, the authors should offer a justification for employing a 2D model.

8.     It is recommended, to illustrate the dam description using a schematic figure that depicts all the dimensions given in lines 129-140.

9.     How was the original section modified in the closure works?

10.  In my view, regarding the technical article, certain sections could benefit from the reduction of information by eliminating redundant data, for instance, the "Location and geology of the Almagrera tailings dam" and "Categories and safety factors" sections.

11.  The parameters or variables outlined in equations (1)-(20) are not clearly defined. It is recommended to provide definitions for all variables at appropriate locations in the text.

12.  A concise description of the derivation process is essential for understanding the yield functions presented in equation (1) and the plastic functions outlined in equation (2).

13.  The data depicted in numerous figures lack standardization. Specifically, it is advisable to employ distinct line types for each selected accelerogram in Fig. 3.

14.  The results and discussion sections also need a thorough improvement. Results are not just about describing the trends and patterns in tables, instead it should be supported by sound scientific intuitions.

15. Irrespective of its merits, the article should undergo meticulous scrutiny by an English language specialist and an expert in the study's field to prevent unnecessary repetition of sentence components, thereby avoiding verbose text.

Comments on the Quality of English Language

Moderate editing of English language required

Author Response

Reviewer 3:

Comments and Suggestions for Authors

This paper presents the dynamic behavior of the dry closure of the Almagrera tailings dam under an earthquake effect. The authors have carried out the investigation via Plaxis 2D program, which uses the finite element method. Three different FE models of unfavorable sections have been studied for the dam body and the tailings. Though the subjected investigation is worthwhile, there are several deficiencies in the manuscript. A major revision of the manuscript is required.

We would like to thank the reviewer for his/her comments. We think that the manuscript has improved. Please, see next the corrections/modifications done accordingly.

  1. Beginning of the abstract need revision.

The Abstract has been rewritten to:

“The Spanish authorities have become more sensitive to the problem of ore tailings following the Aznalcollar dam break. The main novelty of this paper is to study the dynamic behavior for the dry closure of the Almagrera dam under the action of an earthquake. This study has been carried out with the Plaxis 2D program, which uses the finite element (FE) method. The dynamic analysis of the dam has been interpreted in terms of deformations, displacements and principal stresses. Also, the construction of the Uniform Seismic Hazard Acceleration Response Spectrum (USH-ARS) and the selection of real accelerograms for the time-history dynamic calculations is a noted novelty. Numerical analyses show that the dam is safe enough, because a failure surface has not been formed, although several plastic zones may appear in the dam. The FE study of deformations displays that the tailings may attain large deformations, displacements and failure, although this does not jeopardize the safety of the dam, where the displacements are smaller than 3 mm. Nei-ther the tailings nor the dam is expected to suffer liquefaction. It is determined that the 0.09 g threshold value is not exceeded in the acceleration-time graphs on the old reservoir field surface, which is the most critical situation.”

  1. It is recommended to incorporate the critical quantitative results of the study in the abstract.

Some critical and quantitative results have been added to the new abstract.

  1. The novelty of the present study lacks clarity. The authors should explicitly establish, in alignment with the existing technical literature relevant to this paper, the innovative contributions of the article in comparison to other research efforts.

The innovative aspects of this study have been added and explained in the abstract and in the introduction.

  1. It is advisable to avoid the compounding of references.

References have been edited to avoid compounding.

  1. The literature review provided in the introduction section lacks comprehensiveness. It is advisable to critically analyze previous studies in detail, rather than merely summarizing the actions of researchers in the past.

Details of the studies and especially their innovative aspects are presented. Additionally, new studies have been cited to make it more comprehensive.

  1. It is recommended to add recent studies on the dynamic analyses of the rockfill dams using the finite element (FE) method. 

The following text has been added in the Introduction:

“An excellent fit is achieved with displacements and a reasonable one with pore water pressures in a dam using Plaxis 2D [37]. The measurement values obtained with a de-vice and the Plaxis 2D finite element model results were quite consistent. This vali-dates the suitability of using the Plaxis 2D program in similar studies. Numerical analysis of innovative seismic response of earth-rock dams studies were carried out with antiseepage walls [38]. The dynamic viscoelastic constitutive model was utilized in this research. The buckling fracture mechanism of broken rock dam foundation was investigated by gentle transitions and vertical structural discontinuities [39]. The mod-elling philosophy and process, and outcomes for the rock dam foundation are defined and prompted by using numerical methods. A buckling type of failure mechanism is confirmed by analysing the deformation properties resulting from the overloading of the strength reduction of the numerical method. A comprehensive diagnostic method for the safety of tailings dams based on the dynamic weight and quantitative index was studied [40]. For the deformation stability project, the amount and rate of deformation are determined by analysing and interpreting normal operating data in situ ob-servational data and combining them with numerical simulation outcomes.

  1. My main concern regarding the manuscript is that the authors draw conclusions about the structural stability of a highly critical structure based solely on a 2D analysis. In my perspective, the authors should offer a justification for employing a 2D model.

A two-dimensional method is used, but three critical sections are studied, including the maximum section, which is not the one that undergoes the largest permanent deformations. In the literature, it is generally accepted that, for linear structures, a 2D analysis of the most critical sections is an appropriate approximation. Some references from the literature are included as examples.

An excellent fit is achieved with displacements and a reasonable one with pore water pressures in a dam using Plaxis 2D [37]. The measurement values obtained with a de-vice and the Plaxis 2D finite element model results were quite consistent. This vali-dates the suitability of using the Plaxis 2D program in similar studies.

Da Silva, E.M.; Justo, J.L.; Durand, P.; Justo, E.; Vázquez-Boza, M. The Effect of Geotextile Reinforcement and Prefabricated Vertical Drains on the Stability and Settlement of Embankments. Geotext. Geomembranes 2017, 45, 447–461, doi:10.1016/j.geotexmem.2017.07.001.

Chen, D.; Chen, H.; Zhang, W.; Tan, C.; Ma, Z.; Chen, J.; Shan, B. Buckling Failure Mechanism of a Rock Dam Foundation Fractured by Gentle Through-Going and Steep Structural Discontinuities. Sustain. 2020, 12, 1–20, doi:10.3390/su12135426.

A hybrid 1D-2D Lagrangian solver with moving coupling to simulate dam-break flow, Advances in Water Resources 178 (2023) 104487

Performance of 2D-spectral finite element method in dynamic analysis of concrete gravity dams, Structures 59 (2024) 105770

  1. It is recommended, to illustrate the dam description using a schematic figure that depicts all the dimensions given in lines 129-140.

The dimensions and the 3th-5th raisings described have been incorporated to Figure 1.

  1. How was the original section modified in the closure works?

Adding the rockfill reinforcement, downstream, shown in Figure 2.

During the closure works (see Section 3.4), the original section was modified by placing a rockfill reinforcement. In addition, a drawdown was carried out and more material (the Las Viñas fill) was put above the tailings.

  1. In my view, regarding the technical article, certain sections could benefit from the reduction of information by eliminating redundant data, for instance, the "Location and geology of the Almagrera tailings dam" and "Categories and safety factors" sections

The "Location and geology of the Almagrera tailings dam" and "Categories and safety factors" sections have been shortened accordingly.

  1. The parameters or variables outlined in equations (1)-(20) are not clearly defined. It is recommended to provide definitions for all variables at appropriate locations in the text.

The authors have thought about a lot about this reviewer´s comment as we have noticed that there are many equations, that can be hard to follow. However, the aim of this manuscript is not on these (previous and known) equations but in the method followed in this analysis, the seismic hazard, the accelerograms selected and the analysis of the liquefaction.

Therefore, we have decided to add new references for some of these equations (as the authors don´t want to make them the core of this manuscript) and a new Figure for helping to understand Eq. 1-4, which we think that they are the most complicated equations.

A new reference has been added, where the formulation of the Mohr-Coulomb model is described in detail. It is expected to help understanding Eq. 1-4. Also a new figure that shows the Mohr-Coulomb failure surface in principal stress space has been depicted.

New Figure 4. Mohr-Coulomb failure surface in principal stress space

The following text has been added within the manuscript: “The six yield functions (1) describe a hexagonal shape in the stress region [49].

New reference [49].   Amouzou, G.Y.; Soulaïmani, A. Numerical Algorithms for Elastoplacity: Finite Elements Code Development and Implementation of the Mohr–Coulomb Law. Appl. Sci. 2021, 11, 1–27, doi:10.3390/app11104637.

Regarding Equations 5-20, the authors think that they are correctly defined within the text, for example:

“The NCSE-02 method to obtain calculation acceleration is:

                                                                          (5)

where:

- : basic seismic acceleration.

- : dimensionless coefficient which takes into account the return period. For constructions of special importance this is 1.3 -which means a return period of 1,000 years-.

- : soil amplification coefficient.

 =0.803                 for 

: Soil coefficient depending on the classifications of the soil (Table 4).

Table 4. Soil coefficient.

Type of soil

Characteristics

Coefficient

I

Vs > 750 m/s

1.0

II

400 < Vs < 750 m/s

1.3

III

200 < Vs < 400 m/s

1.6

IV

Vs < 200 m/s

2.0

where Vs represents the shear wave velocity with respect to the foundation. For multilayer foundation soils, the first 30 metres should be taken into consideration. In this case,  is calculated as a weighted average:

                                                                                         (6)

where  is the layer thickness.

At the location of the Almagrera dam:

 (Provided by NCSE-02)

 (Special importance, Tr=1,000 years)

  (Basically sound rock)

 So,

The International Commission on Gigantic Dams advises for dams of category C a repetition time of 1000 years and the design acceleration (ad):

ad= 1.3, so, ab= 0.104 g                                                        (7)

The acceleration was embraced in the finite element model. For the design earthquake in dams of category A, this kind of design acceleration was carried out.  For category A dams, the Spanish legislation regards a destructive strong ground motion as well, considered as an excessive movement, by design acceleration:

ad= 2, so, ab= 0.16 g”

  1. A concise description of the derivation process is essential for understanding the yield functions presented in equation (1) and the plastic functions outlined in equation (2).

Please, read Query#11.

  1. The data depicted in numerous figures lack standardization. Specifically, it is advisable to employ distinct line types for each selected accelerogram in Fig. 3.

Figure 3 (now 5) has been modified accordingly. Also, the grid-lines have been reduced to low density grey lines.

New Figure 3 (now 5).

  1. The results and discussion sections also need a thorough improvement. Results are not just about describing the trends and patterns in tables, instead it should be supported by sound scientific intuitions.

The following text has been included:

“4) The expected permanent displacement of the dam body after the maximum ex-pected earthquake is acceptable, with Section 1 being the most critical. The results show that no continuous failure surface crosses the dam body for the maximum ex-pected earthquake. Nonetheless, the results show that, at some short moments the dam body is close to collapse. Moreover, in some places Las Viñas fill and the tailings may attain failure. This indicates a tailings displacement, but not a dam failure. The lique-faction that could suffer the fill upstream is unimportant.”

  1. Irrespective of its merits, the article should undergo meticulous scrutiny by an English language specialist and an expert in the study's field to prevent unnecessary repetition of sentence components, thereby avoiding verbose text.

The whole text has been reviewed by a native English professional reviewer, who has made the necessary corrections.

Reviewer 4 Report

Comments and Suggestions for Authors

The purpose of this paper is to investigate the dynamic behaviour of the dry closure of the Almagrera tailings dam under an earthquake effect.  Real accelerograms have been selected by means of earthquake hazard and risk acceleration response spectrums in Almagrera for dynamic analyses. Numerical analyses results show that the dam is safe enough. The research is meaningful, and before it is accepted, the authors should optimize and modify it according to the following comments:

1.     In the introduction section, can the authors summarize what software and models have been used in previous studies on Dynamic Analysis, which is more relevant to the research in this article.

2.     In the Location and geography of the Almagerera tails dam, the authors provided a sectional view, but readers are not familiar with the layout of the tails dam. Can you provide it? If it is not possible, a corresponding description should be provided.

3.     In Chapter 3, the author provides multiple phases and corresponding steps, which can provide a flowchart for more convenient expression.

4.     Suggest the authors to explain what parameters were used to input Seismic and Seismic input data, what were they, which ones were used in your study, and why?

5.     For Equivalent layers for sections 1-2-3, the authors have formed a table that is good, but adding corresponding figures may make it clearer.

6.     Figure 8 is too unclear. How it was formed? And it should be provided as a clearer vector diagram.

7.     Suggest simplifying the conclusion and increasing the sense of hierarchy. Currently, there is only one paragraph, making it difficult for readers to read the most important content.

8.     Additionally, some formatting errors, such as superscripts of squares or cubes; The beginning of a paragraph should be two characters empty. Some fonts are large, while others are small. Some Figures are placed in the middle of a paragraph. Also, please verify if the reference starts with the subject of a sentence, is it should be in the current format?

Comments on the Quality of English Language

Read the entire text thoroughly to avoid any expression and grammar errors.

Author Response

Reviewer 4:

Comments and Suggestions for Authors

The purpose of this paper is to investigate the dynamic behaviour of the dry closure of the Almagrera tailings dam under an earthquake effect.  Real accelerograms have been selected by means of earthquake hazard and risk acceleration response spectrums in Almagrera for dynamic analyses. Numerical analyses results show that the dam is safe enough. The research is meaningful, and before it is accepted, the authors should optimize and modify it according to the following comments:

We would like to thank the reviewer for his/her comments and cheers. We think that the manuscript has improved. Please, see next the corrections/modifications done accordingly.

  1. In the introduction section, can the authors summarize what software and models have been used in previous studies on Dynamic Analysis, which is more relevant to the research in this article.

In the introduction section, additional references are given by the authors about which software and models were used in previous studies on Dynamic Analysis, which are more relevant to the research in this article.

An excellent fit is achieved with displacements and a reasonable one with pore water pressures in a dam using Plaxis 2D [37]. The measurement values obtained with a de-vice and the Plaxis 2D finite element model results were quite consistent. This vali-dates the suitability of using the Plaxis 2D program in similar studies. Numerical analysis of innovative seismic response of earth-rock dams studies were carried out with antiseepage walls [38]. The dynamic viscoelastic constitutive model was utilized in this research. The buckling fracture mechanism of broken rock dam foundation was investigated by gentle transitions and vertical structural discontinuities [39]. The modelling philosophy and process, and outcomes for the rock dam foundation are defined and prompted by using numerical methods. A buckling type of failure mechanism is confirmed by analysing the deformation properties resulting from the overloading of the strength reduction of the numerical method. A comprehensive diagnostic method for the safety of tailings dams based on the dynamic weight and quantitative index was studied [40]. For the deformation stability project, the amount and rate of deformation are determined by analysing and interpreting normal operating data in situ observational data and combining them with numerical simulation outcomes.

Finally, the dynamic calculation has been performed with Plaxis 2D software and the results interpreted (these have been added to the Introduction Section).

  1. In the Location and geography of the Almagerera tails dam, the authors provided a sectional view, but readers are not familiar with the layout of the tails dam. Can you provide it? If it is not possible, a corresponding description should be provided.

The products of the excavations that are obtained are placed in the dam, but with rockfill at the base. There are also different drains (in yellow) and quarry-run (permeable) on the outside.

This text has been added within the text of the manuscript.

  1. In Chapter 3, the author provides multiple phases and corresponding steps, which can provide a flowchart for more convenient expression.

A flowchart has been added.

  1. Suggest the authors to explain what parameters were used to input Seismic and Seismic input data, what were they, which ones were used in your study, and why?

Please see Section 4.3 and 4.4:

“4.3. Spanish regulation

The Spanish regulation that states the criteria and calculation methods for seismicity is called Norma Sismorresistente, 2002 –Earthquake-resistant Construction Regulations- (NCSE-02). This regulation provides a seismic hazard map with basic acceleration ( ) isolines. In addition, it supplies a list of the basic seismic accelerations for every town or city. For Calañas village  =0.08g.

According to the regulation, the seismicity for this basic acceleration is:

Medium: 0.04g< <0.13g. Pseudo-statics methods can be used. Technical Guides (2 and 3) of the Spanish Committee on Large Dams (SPANCOLD) consider constant forces acting on the centre of gravity. The horizontal component is produced by the basic acceleration; the vertical one is reduced by 0.7. Basic seismic acceleration has to be modified depending on the specific case. The NCSE-02 method to obtain calculation acceleration is:

                                                                         (5)

where:

- : basic seismic acceleration.

- : dimensionless coefficient which takes into account the return period. For constructions of special importance this is 1.3 -which means a return period of 1,000 years-.

- : soil amplification coefficient.

 =0.803                 for 

: Soil coefficient depending on the classifications of the soil (Table 4).

Table 4. Soil coefficient.

Type of soil

Characteristics

Coefficient

I

Vs > 750 m/s

1.0

II

400 < Vs < 750 m/s

1.3

III

200 < Vs < 400 m/s

1.6

IV

Vs < 200 m/s

2.0

where Vs represents the shear wave velocity with respect to the foundation. For multilayer foundation soils, the first 30 metres should be taken into consideration. In this case,  is calculated as a weighted average:

                                                                             (6)

where  is the layer thickness.

At the location of the Almagrera dam:

 (Provided by NCSE-02)

 (Special importance, Tr=1,000 years)

  (Basically sound rock)

 So,

The International Commission on Gigantic Dams advises for dams of Category C a repetition time of 1000 years and the design acceleration (ad):

ad= 1.3, so, ab= 0.104 g                                                       (7)

The acceleration was embraced in the finite element model. For the design earthquake in dams of Category A, this kind of design acceleration was carried out.  For Category A dams, the Spanish legislation contemplates a destructive strong ground motion as well, considered as an excessive movement, by design acceleration:

ad= 2, so, ab= 0.16 g                                      (8)

Regarding Table 2, seismic forces should be calculated as accidental actions; therefore, the safety factor should be at least 1.2. The consequences of the pseudo-static calculations are listed in Table 3. The lowest safety factor is 1.34, obtained for the drawdown phase (>1.2). A safety factor of 1.03 has been calculated inside the tailings -not in the dam- and for this reason the dam is considered safe enough. It is important to note that a dynamic calculation for Almagrera is not required by the Spanish regulation. However, considering that the Almagrera tailings dam includes the largest mine waste deposit in the Andalusian region makes a dynamic calculation appealing. Large displacements occur inside the tailings.

Later, in Section 4.4

“4.4. Definition of the accelerograms

The initial step in a seismic analysis is the definition of the accelerograms that will be applied. Where many accelerograms have been recorded for years, accelerograms near the site could be used. By contrast, where there are not so many data available, using visco-elastic response spectra is usually a better option. The probabilistic method for choosing calculation accelerograms proposed by [56] has been used. This method is grounded on the construction of a USHARS for the site, considering the kind of ground and the desired risk situation –time of exposure and possibility of exceeding it. Then, the USHARS is compared with the acceleration response spectrum of real accelerograms recorded in the identical kind of ground. Finally, the accelerograms with a scale factor (f) close to 1 (Table 5) and a minor standard declination have been chosen.

Table 5. Selected accelerograms for the Almagrera mine waste fill dam.

Accelerograms

f

358

1.08

385

0.915

607

1.043

4341

1.006

6261

1.050

6269

0.964

6274

0.923

For the Almagrera location, the following features and parameters have been considered:

- Founded on sound rock.

- Repetition time shut to 1,000 years.

Real accelerograms were obtained at http://www.isesd.hi.is/. Data is available in the European Strong Movement Database, available online at this online address. The accelerograms selected are given in Table 6. Figure 5 displays the USHARS for the site and compares it with the spectrum of selected earthquake records.

  1. For Equivalent layers for sections 1-2-3, the authors have formed a table that is good, but adding corresponding figures may make it clearer.

The corresponding figure has been added in the text.

Figure 6. FE model for the three sections.

  1. Figure 8 is too unclear. How was it formed? And it should be provided as a clearer vector diagram.

The Figure 8 has been modified: The ore tailings area has been shortened and the dam enlarged. Please, note that also Figure 1 has been improved, showing from the 3rd to the 5th phase of the dam.

  1. Suggest simplifying the conclusion and increasing the sense of hierarchy. Currently, there is only one paragraph, making it difficult for readers to read the most important content.

The conclusion section has been modified according to the reviewer´s suggestion.

  1. Additionally, some formatting errors, such as superscripts of squares or cubes; The beginning of a paragraph should be two characters empty. Some fonts are large, while others are small. Some Figures are placed in the middle of a paragraph. Also, please verify if the reference starts with the subject of a sentence, is it should be in the current format?

The formatting errors have been corrected.

Round 2

Reviewer 1 Report

Comments and Suggestions for Authors

acceptance

Comments on the Quality of English Language

acceptance

Reviewer 2 Report

Comments and Suggestions for Authors

I thank the authors for their commitment to corrections based on the recommendations made and hope that this work will be useful for future development work.

Reviewer 3 Report

Comments and Suggestions for Authors

The manuscript has been enhanced by including all of the comments and suggestions. 

Reviewer 4 Report

Comments and Suggestions for Authors

Thank you for the author's revisions to the paper. I personally believe it is acceptable.